# Out-of-distribution Detection Learning with Unreliable Out-of-distribution Sources

**Haotian Zheng**[1,2*]  **Qizhou Wang**[1*]  **Zhen Fang**[3]  **Xiaobo Xia**[4]
**Feng Liu**[5]  **Tongliang Liu**[4]  **Bo Han**[1†]

[1]Department of Computer Science, Hong Kong Baptist University
[2]School of Electronic Engineering, Xidian University
[3]Australian Artificial Intelligence Institute, University of Technology Sydney
[4]Sydney AI Centre, The University of Sydney
[5]School of Computing and Information Systems, The University of Melbourne
htzheng.xdu@gmail.com    {csqzwang, bhanml}@comp.hkbu.edu.hk
zhen.fang@uts.edu.au    xiaoboxia.uni@gmail.com
fengliu.ml@gmail.com    tongliang.liu@sydney.edu.au

## Abstract

Out-of-distribution (OOD) detection discerns OOD data where the predictor cannot make valid predictions as in-distribution (ID) data, thereby increasing the reliability of open-world classification. However, it is typically hard to collect real out-of-distribution (OOD) data for training a predictor capable of discerning ID and OOD patterns. This obstacle gives rise to *data generation-based learning methods*, synthesizing OOD data via data generators for predictor training without requiring any real OOD data. Related methods typically pre-train a generator on ID data and adopt various selection procedures to find those data likely to be the OOD cases. However, generated data may still coincide with ID semantics, i.e., mistaken OOD generation remains, confusing the predictor between ID and OOD data. To this end, we suggest that generated data (with mistaken OOD generation) can be used to devise an *auxiliary OOD detection task* to facilitate real OOD detection. Specifically, we can ensure that learning from such an auxiliary task is beneficial if the ID and the OOD parts have disjoint supports, with the help of a well-designed training procedure for the predictor. Accordingly, we propose a powerful data generation-based learning method named *Auxiliary Task-based OOD Learning* (ATOL) that can relieve the mistaken OOD generation. We conduct extensive experiments under various OOD detection setups, demonstrating the effectiveness of our method against its advanced counterparts. The code is publicly available at: https://github.com/tmlr-group/ATOL.

## 1  Introduction

Deep learning in the open world should not only make accurate predictions for in-distribution (ID) data meanwhile should detect out-of-distribution (OOD) data whose semantics are different from ID cases [15, 56, 22, 81, 85, 82]. It drives recent studies in OOD detection [43, 23, 33, 66, 80, 24], which is important for many safety-critical applications such as autonomous driving and medical analysis. Previous works have demonstrated that well-trained predictors can wrongly take many OOD data as ID cases [1, 15, 21, 48], motivating recent studies towards effective OOD detection.

---

[*]Equal contributions.
[†]Correspondence to Bo Han (bhanml@comp.hkbu.edu.hk).

37th Conference on Neural Information Processing Systems (NeurIPS 2023).

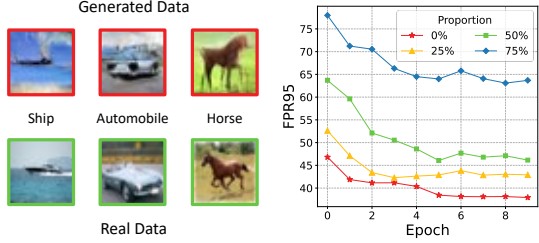
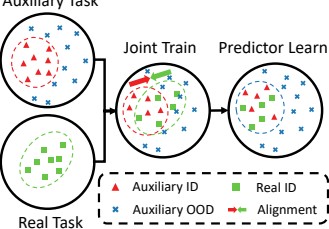

(a) Mistaken Semantics  (b) Adverse Impacts

Figure 1: Illustrations of mistaken OOD generation. Figure 1(a) indicates the generated OOD data (colored in red) may wrongly possess semantics of the real ID data (colored in green), e.g., generated data with the ID semantics of "Horse". Figure 1(b) demonstrates that the OOD performance of predictors are impaired by unreliable OOD data, described in Appendix E.2. With the proportion of OOD data possessing ID semantics increasing, the OOD performance (measured by FPR95) degrades.

Figure 2: Illustration of our method. ATOL makes the predictor learn from the real task and our auxiliary task jointly via a well-designed learning procedure, leading to the separation between the OOD and ID cases. The predictor can benefit from such an auxiliary task for real OOD detection, relieving mistaken OOD generation.

In the literature, an effective learning scheme is to conduct model regularization with OOD data, making predictors achieve low-confidence predictions on such data [3, 16, 36, 42, 49, 75, 79]. Overall, they directly let the predictor learn to discern ID and OOD patterns, thus leading to improved performance in OOD detection. However, it is generally hard to access real OOD data [10, 11, 69, 70], hindering the practical usage scenarios for such a promising learning scheme.

Instead of collecting real OOD data, we can generate OOD data via data generators to benefit the predictor learning, motivating the *data generation-based methods* [31, 50, 55, 62, 67]. Generally speaking, existing works typically fit a data generator [12] on ID data (since real OOD data are inaccessible). Then, these strategies target on finding the OOD-like cases from generated data by adopting various selection procedures. For instance, Lee et al. [31] and Vernekar et al. [62] take boundary data that lie far from ID boundaries as OOD-like data; PourReza et al. [50] select those data generated in the early stage of generator training. However, since the generators are trained on ID data and these selection procedures can make mistakes, one may wrongly select data with ID semantics as OOD cases (cf., Figure 1(a)), i.e., *mistaken OOD generation* remains. It will misguide the predictors to confuse between ID and OOD data, making the detection results unreliable, as in Figure 1(b). Overall, it is hard to devise generator training procedures or data selection strategies to overcome mistaken OOD generation, thus hindering practical usages of previous data generation-based methods.

To this end, we suggest that generated data can be used to devise an *auxiliary task*, which can benefit real OOD detection even with data suffered from mistaken OOD generation. Specifically, such an auxiliary task is a crafted OOD detection task, thus containing ID and OOD parts of data (termed *auxiliary ID* and *auxiliary OOD data*, respectively). Then, two critical problems arise —*how to make such an auxiliary OOD detection task learnable w.r.t. the predictor* and *how to ensure that the auxiliary OOD detection task is beneficial w.r.t. the real OOD detection*. For the first question, we refer to the advanced OOD detection theories [11], suggesting that the auxiliary ID and the auxiliary OOD data should have the disjoint supports in the input space (i.e., without overlap w.r.t. ID and OOD distributions), cf., **C**1 (Condition 1 for short). For the second question, we justify that, for our predictor, if real and auxiliary ID data follow similar distributions, cf., **C**2, the auxiliary OOD data are reliable OOD data w.r.t. the real ID data. In summary, if **C**1 and **C**2 hold, we can ensure that learning from the auxiliary OOD detection task can benefit the real OOD detection, cf., Proposition 1.

Based on the auxiliary task, we propose *Auxiliary Task-based OOD Learning* (ATOL), an effective data generation-based OOD learning method. For **C**1, it is generally hard to directly ensure the disjoint supports between the auxiliary ID and auxiliary OOD data due to the high-dimensional input space. Instead, we manually craft two disjoint regions in the low-dimensional latent space (i.e., the input space of the generator). Then, with the generator having *distance correlation* between input

space and latent space [63], we can make the generated data that belong to two different regions in the latent space have the disjoint supports in the input space, assigning to the auxiliary ID and the auxiliary OOD parts, respectively. Furthermore, to fulfill **C**2, we propose a distribution alignment risk for the auxiliary and the real ID data, alongside the OOD detection risk to make the predictor benefit from the auxiliary OOD detection task. Following the above learning procedure, we can relieve the mistaken OOD generation issue and achieve improved performance over previous works in data generation-based OOD detection, cf., Figure 2 for a heuristic illustration.

To verify the effectiveness of ATOL, we conduct extensive experiments across representative OOD detection setups, demonstrating the superiority of our method over previous data generation-based methods, such as Vernekar et al. [62], with $12.02\%$, $18.32\%$, and $20.70\%$ improvements measured by average FPR95 on CIFAR-10, CIFAR-100, and ImageNet datasets. Moreover, compared with more advanced OOD detection methods (cf., Appendix F), such as Sun et al. [57], our ATOL can also have promising improvements, which reduce the average FPR95 by $4.36\%$, $15.03\%$, and $12.02\%$ on CIFAR-10, CIFAR-100, and ImageNet datasets.

**Contributions.** We summarize our main contributions into three folds:

- We focus on the mistaken OOD generation in data generation-based detection methods, which are largely overlooked in previous works. To this end, we introduce an auxiliary OOD detection task to combat the mistaken OOD generation and suggest a set of conditions in Section 3 to make such an auxiliary task useful.

- Our discussion about the auxiliary task leads to a practical learning method named ATOL. Over existing works in data generation-based methods, our ATOL introduces small costs of extra computations but is less susceptible to mistaken OOD generation.

- We conduct extensive experiments across representative OOD detection setups, demonstrating the superiority of our method over previous data generation-based methods. Our ATOL is also competitive over advanced works that use real OOD data for training, such as outlier exposure [16], verifying that the data generation-based methods are still worth studying.

## 2 Preliminary

Let $\mathcal{X} \subseteq \mathbb{R}^n$ the input space, $\mathcal{Y} = \{1, \ldots, c\}$ the label space, and $\mathbf{h} = \boldsymbol{\rho} \circ \boldsymbol{\phi} : \mathbb{R}^n \to \mathbb{R}^c$ the predictor, where $\boldsymbol{\phi} : \mathbb{R}^n \to \mathbb{R}^d$ is the feature extractor and $\boldsymbol{\rho} : \mathbb{R}^d \to \mathbb{R}^c$ is the classifier. We consider $\mathcal{P}_{X,Y}^{\text{ID}}$ and $\mathcal{P}_X^{\text{ID}}$ the joint and the marginal ID distribution defined over $\mathcal{X} \times \mathcal{Y}$ and $\mathcal{X}$, respectively. We also denote $\mathcal{P}_X^{\text{OOD}}$ the marginal OOD distribution over $\mathcal{X}$. Then, the main goal of OOD detection is to find the proper predictor $\mathbf{h}(\cdot)$ and the scoring function $s(\cdot; \mathbf{h}) : \mathbb{R}^n \to \mathbb{R}$ such that the OOD detector

$$f_\beta(\mathbf{x}) = \left\{ \begin{array}{ll} \text{ID}, & \text{if } s(\mathbf{x}; \mathbf{h}) \geq \beta, \\ \text{OOD}, & \text{if } s(\mathbf{x}; \mathbf{h}) < \beta, \end{array} \right. \tag{1}$$

can detect OOD data with a high success rate, where $\beta$ is a given threshold. To get a proper predictor $\mathbf{h}(\cdot)$, outlier exposure [16] proposes regularizing the predictor to produce low-confidence predictions for OOD data. Assuming the real ID data $\mathbf{x}_{\text{ID}}$ and label $y_{\text{ID}}$ are randomly drawn from $\mathcal{P}_{X,Y}^{\text{ID}}$, then the learning objective of outlier exposure can be written as:

$$\min_{\mathbf{h}} \mathbb{E}_{\mathcal{P}_{X,Y}^{\text{ID}}} \left[ \ell_{\text{CE}}(\mathbf{x}_{\text{ID}}, y_{\text{ID}}; \mathbf{h}) \right] + \lambda \mathbb{E}_{\mathcal{P}_X^{\text{OOD}}} \left[ \ell_{\text{OE}}(\mathbf{x}; \mathbf{h}) \right], \tag{2}$$

where $\lambda$ is the trade-off hyper-parameter, $\ell_{\text{CE}}(\cdot)$ is the cross-entropy loss, and $\ell_{\text{OE}}(\cdot)$ is the Kullback-Leibler divergence of softmax predictions to the uniform distribution. Although outlier exposure remains one of the most powerful methods in OOD detection, its critical reliance on OOD data hinders its practical applications.

Fortunately, data generation-based OOD detection methods can overcome the reliance on real OOD data, meanwhile making the predictor learn to discern ID and OOD data. Overall, existing works [31, 62, 67] leverage the *generative adversarial network* (GAN) [12, 51] to generate data, collecting those data that are likely to be the OOD cases for predictor training. As a milestone work, Lee et al. [31] propose selecting the "boundary" data that lie in the low-density area of $\mathcal{P}_X^{\text{ID}}$, with the following learning objective for OOD generation:

$$\min_G \max_D \underbrace{\mathbb{E}_{\mathcal{G}_X} \left[ \ell_{\text{OE}}(\mathbf{x}; \mathbf{h}) \right]}_{\text{(a) generating boundary data}} + \underbrace{\mathbb{E}_{\mathcal{P}_X^{\text{ID}}} [\log D(\mathbf{x}_{\text{ID}})] + \mathbb{E}_{\mathcal{G}_X} [\log(1 - D(\mathbf{x}))]}_{\text{(b) training data generator}}, \tag{3}$$

where $\mathcal{G}_X$ denotes the OOD distribution of the generated data w.r.t. the generator $G : \mathbb{R}^m \to \mathbb{R}^n$ ($m$ is the dimension of the latent space). Note that the term (a) forces the generator $G(\cdot)$ to generate low-density data and the term (b) is the standard learning objective for GAN training [12]. Then, with the generator trained for OOD generation, the generated data are used for predictor training, following the outlier exposure objective, given by

$$\min_{\mathbf{h}} \mathbb{E}_{\mathcal{P}^{\text{ID}}_{X,Y}} [\ell_{\text{CE}}(\mathbf{x}_{\text{ID}}, y_{\text{ID}}; \mathbf{h})] + \lambda \mathbb{E}_{\mathcal{G}_X} [\ell_{\text{OE}}(\mathbf{x}; \mathbf{h})]. \tag{4}$$

**Drawbacks.** Although a promising line of works, the generated data therein may still contain many ID semantics (cf., Figure 1(a)). It stems from the lack of knowledge about OOD data, where one can only access ID data to guide the training of generators. Such mistaken OOD generation is common in practice, and its negative impacts are inevitable in misleading the predictor (cf., Figure 1(b)). Therefore, how to overcome the mistaken OOD generation is one of the key challenges in data generation-based OOD detection methods.

## 3  Motivation

Note that getting generators to generate reliable OOD data is difficult, since there is no access to real OOD data in advance and the selection procedures can make mistakes. Instead, given that generators suffer from mistaken OOD generation, we aim to make the predictors alleviate the negative impacts raised by those unreliable OOD data.

Our key insight is that these unreliable generated data can be used to devise a reliable *auxiliary task* which can benefit the predictor in real OOD detection. In general, such an auxiliary task is a crafted OOD detection task, of which the generated data therein are separated into the ID and the OOD parts. Ideally, the predictor can learn from such an auxiliary task and *transfer* the learned knowledge (from the auxiliary task) to benefit the real OOD detection, thereafter relieving the mistaken OOD generation issue. Therein, two issues require our further study:

- Based on unreliable OOD sources, how to devise a learnable auxiliary OOD detection task?
- Given the well-designed auxiliary task, how to make it benefit the real OOD detection?

For the first question, we refer the advanced work [11] in OOD detection theories, which provides necessary conditions for which the OOD detection task is learnable. In the context of the auxiliary OOD detection task, we will separate the generated data into the *auxiliary ID* parts and *auxiliary OOD* parts, following the distributions defined by $\mathcal{G}_X^{\text{ID}}$ and $\mathcal{G}_X^{\text{OOD}}$, respectively. Then, we provide the following condition for the *separable property* of the auxiliary OOD detection task.

**Condition 1** (**Separation for the Auxiliary Task**). *There is no overlap between the auxiliary ID distribution and the auxiliary OOD distribution, i.e., $\text{supp}\,\mathcal{G}_X^{\text{ID}} \cap \text{supp}\,\mathcal{G}_X^{\text{OOD}} = \emptyset$, where $\text{supp}$ denotes the support set of a distribution.*

Overall, **C**1 states that the auxiliary ID data distribution $\mathcal{G}_X^{\text{ID}}$ and the auxiliary OOD data distribution $\mathcal{G}_X^{\text{OOD}}$ should have the disjoint supports. Then, according to Fang et al. [11] (Theorem 3), such an auxiliary OOD detection might be learnable, in that the predictor can have a high success rate in discerning the auxiliary ID and the auxiliary OOD data. In general, the separation condition is indispensable for us to craft a valid OOD detection task. Also, as in the proof of Proposition 1, the separation condition is also important to benefit the real OOD detection.

For the second question, we need to first formally define if a data point can benefit the real OOD detection. Here, we generalize the separation condition in Fang et al. [11], leading to the following definition for the *reliability* of OOD data.

**Definition 1** (**Reliability of OOD Data**). *An OOD data point $\mathbf{x}$ is reliable (w.r.t. real ID data) to a mapping function $\phi'(\cdot)$, if $\phi'(\mathbf{x}) \notin \text{supp}\,\phi'_{\#}\mathcal{P}_X^{\text{ID}}$, where $\phi'_{\#}$ is the distribution transformation associated with $\phi'(\cdot)$.*

With identity mapping function $\phi'(\cdot)$, the above definition states that $\mathbf{x}$ is a reliable OOD data point if it is not in the support set of the real ID distribution w.r.t. the transformed space. Intuitively, if real ID data are representative almost surely [11], $\phi'(\cdot)$ far from the ID support also indicates that $\phi'(\cdot)$ will not possess ID semantics, namely, reliable. Furthermore, $\phi'(\cdot)$ is not limited to identity mapping,

which can be defined by any transformed space. The feasibility of such a generalized definition is supported by many previous works [10, 41] in OOD learning, demonstrating that reliable OOD data defined in the embedding space (or in our context, the transformed distribution) can also benefit the predictor learning in OOD detection.

Based on Definition 1, we want to make auxiliary OOD data reliable w.r.t. real ID data, such that the auxiliary OOD detection task is ensured to be beneficial, which motivates the following condition.

**Condition 2** (**Transferability of the Auxiliary Task**). *The auxiliary OOD detection task is transferable w.r.t. the real OOD detection task if $\phi'_\# \mathcal{P}^{ID}_X \approx \phi'_\# \mathcal{G}^{ID}_X$.*

Overall, **C**2 states that auxiliary ID data approximately follow the same distribution as that of the real ID data in the transformed space, i.e., $\phi'(\mathcal{X})$. Note that, Auxiliary ID and OOD data can arbitrarily differ from real ID and OOD data in the data space, i.e., *auxiliary data are unreliable OOD sources*. However, if **C**2 is satisfied, the model "*believes*" the auxiliary ID data and the real ID data are the same. Then, we are ready to present our main proposition, demonstrating why **C**2 can ensure the reliability of auxiliary OOD data.

**Proposition 1.** *Assume the predictor* $\mathbf{h} = \boldsymbol{\rho} \circ \boldsymbol{\phi}$ *can separate ID and OOD data, namely,*

$$\mathbb{E}_{\mathcal{G}^{ID}_{X,Y}}[\ell_{CE}(\mathbf{x}_{ID}, y_{ID}; \mathbf{h})] + \lambda \mathbb{E}_{\mathcal{G}^{OOD}_X}[\ell_{OE}(\mathbf{x}; \mathbf{h})] \tag{5}$$

*approaches 0. Under* **C**1 *and* **C**2*, auxiliary OOD data are reliable w.r.t. real ID data given* $\phi' = \phi$.

The proof can be found in Appendix C. In heuristics, Eq. (5) states that the predictor should excel at such an auxiliary OOD detection task, where the model can almost surely discern the auxiliary ID and the auxiliary OOD cases. Then, if we can further align the real and the auxiliary ID distributions (i.e., **C**2), the predictor will make no difference between the real ID data and the auxiliary ID data. Accordingly, since the auxiliary OOD data are reliable OOD cases w.r.t. the auxiliary ID data, the auxiliary OOD data are also reliable w.r.t. the real ID data. Thereafter, one can make the predictor learn to discern the real ID and auxiliary OOD data, where no mistaken OOD generation affects. Figure 2 summarizes our key concepts. As we can see, learning from the auxiliary OOD detection task and further aligning the ID distribution ensure the separation between ID and OOD cases, benefiting the predictor learning from reliable OOD data and improving real OOD detection performance.

**Remark 1.** *There are two parallel definitions for real OOD data [70], related to different goals towards effective OOD detection. One definition is that an exact distribution of real OOD data exists, and the goal is to mitigate the distribution discrepancy w.r.t. the real OOD distribution for effective OOD detection. Another definition is that all those data whose true labels are not in the considered label space are real OOD data, and the goal is to make the predictor learn to see as many OOD cases as possible. Due to the lack of real OOD data for supervision, the latter definition is more suitable than the former one in our context, where we can concentrate on mistaken OOD generation.*

## 4 Learning Method

Section 3 motivates a new data generation-based method named *Auxiliary Task-based OOD Learning* (ATOL), overcoming the mistaken OOD generation via the auxiliary OOD detection task. Referring to Theorem 1, the power of the auxiliary task is built upon **C**1 and **C**2. It makes ATOL a two-staged learning scheme, related to auxiliary task crafting (crafting the auxiliary task for **C**1) and predictor training (applying the auxiliary task for **C**2), respectively. Here, we provide the detailed discussion.

### 4.1 Crafting the Auxiliary Task

We first need to construct the auxiliary OOD detection task, defined by auxiliary ID and auxiliary OOD data generated by the generator $G(\cdot)$. In the following, we assume the auxiliary ID data $\hat{\mathbf{x}}_{ID}$ drawn from $\mathcal{G}^{ID}_X$ and the auxiliary OOD data $\hat{\mathbf{x}}_{OOD}$ drawn from $\mathcal{G}^{OOD}_X$. Then, according to **C**1, we need to fulfill the condition that $\mathcal{G}^{ID}_X$ and $\mathcal{G}^{OOD}_X$ should have the disjoint supports.

Although all distribution pairs $(\mathcal{G}^{ID}_X, \mathcal{G}^{OOD}_X)$ with disjoint supports can make **C**1 hold, it is hard to realize such a goal due to the complex data space $\mathbb{R}^n$ of images. Instead, we suggest crafting such a distribution pair in the latent space $\mathbb{R}^m$ of the generator, of which the dimension $m$ is much lower

than $n$. We define the latent ID and the latent OOD data as $\mathbf{z}_{\text{ID}}$ and $\mathbf{z}_{\text{OOD}}$, respectively. Then, we assume the latent ID data are drawn from the high-density region of a Mixture of Gaussian (MoG):

$$\mathbf{z}_{\text{ID}} \in \mathcal{Z}^{\text{ID}} \text{ with } \mathcal{Z}^{\text{ID}} = \{\mathbf{z} \sim \mathcal{M}_Z | \mathcal{M}(\mathbf{z}) > \tau\}, \tag{6}$$

where $\tau$ is the threshold and $\mathcal{M}_Z$ is a pre-defined MoG, given by the density function of MoG $\mathcal{M}(\mathbf{z}) = \sum_{i=1}^{c} \frac{1}{c} \mathcal{N}(\mathbf{z} | \boldsymbol{\mu}_i, \boldsymbol{\sigma}_i)$, with the mean $\boldsymbol{\mu}_i$ and the covariance $\boldsymbol{\sigma}_i$ for the $i$-th sub-Gaussian. Furthermore, as we demonstrate in Section 4.2, we also require the specification of labels for latent ID data. Therefore, we assume that $\mathcal{M}_Z$ is constructed by $c$ sub-distributions of Gaussian, and data belong to each of these $c$ sub-distributions should have the same label $\hat{y}_{\text{ID}}$.

**Remark 2.** *MoG provides a simple way to generate data, yet the key point is to ensure that auxiliary ID/OOD data should have the disjoint support, i.e, **C**1. Therefore, if **C**1 is satisfied properly, other noise distributions, such as the beta mixture models[40] and the uniform distribution, can also be used. Further, our ATOL is different from previous data generation-based methods in that we do not require generated data to be reliable in the data space. ATOL does not involve fitting the MoG to real ID data, where the parameters can be pre-defined and fixed. Therefore, we do not consider overfitting and accuracy of MoG in our paper.*

Furthermore, for the disjoint support, the latent OOD data are drawn from a uniform distribution except for the region with high MoG density, i.e.,

$$\mathbf{z}_{\text{OOD}} \in \mathcal{Z}^{\text{OOD}} \text{ with } \mathcal{Z}^{\text{OOD}} = \{\mathbf{z} \sim \mathcal{U}_Z | \mathcal{M}(\mathbf{z}) \leq \tau\}, \tag{7}$$

with $\mathcal{U}_Z$ a pre-defined uniform distribution. For now, we can ensure that $\mathcal{Z}^{\text{ID}} \cap \mathcal{Z}^{\text{OOD}} = \emptyset$.

However, $\mathcal{Z}^{\text{ID}} \cap \mathcal{Z}^{\text{OOD}} = \emptyset$ does not imply $G(\mathcal{Z}^{\text{ID}}) \cap G(\mathcal{Z}^{\text{OOD}}) = \emptyset$, i.e., **C**1 is not satisfied given arbitrary $G(\cdot)$. To this end, we suggest further ensuring the generator to be a distance-preserving function [83], which is a sufficient condition to ensure that the disjoint property can be transformed from the latent space into the data space. For distance preservation, we suggest a regularization term for the generator regularizing, which is given by

$$\ell_{\text{reg}}(\mathbf{z_1}, \mathbf{z_2}; G) = -\frac{\mathbb{E}_{\mathcal{U}_Z \times \mathcal{U}_Z}[d(\mathbf{z}_1, \mathbf{z}_2) \cdot d(G(\mathbf{z}_1), G(\mathbf{z}_2))]}{\sqrt{\mathbb{E}_{\mathcal{U}_Z \times \mathcal{U}_Z}[d(\mathbf{z}_1, \mathbf{z}_2)] \, \mathbb{E}_{\mathcal{U}_Z \times \mathcal{U}_Z}[d(G(\mathbf{z}_1), G(\mathbf{z}_2))]}}, \tag{8}$$

where $d(\mathbf{z}_1, \mathbf{z}_2)$ is the centralized distance given by $\|\mathbf{z}_1 - \mathbf{z}_2\|_2 - \mathbb{E}\|\mathbf{z}_1 - \mathbf{z}_2\|_2$. Overall, Eq. (8) is the correlation for the distances between data pairs measured in the latent and the data space, stating that the closer distances in the latent space should indicate the closer distances in the data space.

In summary, to fulfill **C**1, we need to specify the regularizing procedure for the generator, overall summarized by the following two steps. First, we regularize the generator with the regularization in Eq. (8) to ensure the distance-preserving property. Then, we specify two disjoint regions in the latent space following Eqs. (6)-(7), of which the data after passing through the generator are taken as auxiliary ID and auxiliary OOD data, respectively. We also summarize the algorithm details in crafting the auxiliary OOD detection task in Appendix D.

## 4.2 Applying the Auxiliary Task

We have discussed a general strategy to construct the auxiliary OOD detection task, which can be learned by the predictor following an outlier exposure-based learning objective, similar to Eq. (2). Now, we fulfill **C**2, transferring model capability from the auxiliary task to real OOD detection.

In general, **C**2 states that we should align distributions between the auxiliary and the real ID data such that the auxiliary OOD data can benefit real OOD detection, thus overcoming the negative effects of mistaken OOD generation. Following Tang et al. [59], we align the auxiliary and the real ID distribution via supervised contrastive learning [29], pulling together data belonging to the same class in the embedding space meanwhile pushing apart data from different classes. Accordingly, we suggest the alignment loss for the auxiliary ID data $\hat{\mathbf{x}}_{\text{ID}}$, following,

$$\ell_{\text{align}}(\hat{\mathbf{x}}_{\text{ID}}, \hat{y}_{\text{ID}}; \phi) = -\log \frac{1}{|\mathcal{N}_l^{\hat{y}_{\text{ID}}}|} \sum_{\mathbf{x}_{\text{ID}}^p \in \mathcal{N}_l^{\hat{y}_{\text{ID}}}} \frac{\exp[\phi(\hat{\mathbf{x}}_{\text{ID}}) \cdot \phi(\mathbf{x}_{\text{ID}}^p)]}{\sum_{\mathbf{x}_{\text{ID}}^a \in \mathcal{N}_l} \exp[\phi(\hat{\mathbf{x}}_{\text{ID}}) \cdot \phi(\mathbf{x}_{\text{ID}}^a)]}. \tag{9}$$

$\mathcal{N}_l$ is a set of data of size $l$ drawn from the real ID distribution, namely, $\mathcal{N}_l = \{\mathbf{x}_{\text{ID}}^i | (\mathbf{x}_{\text{ID}}^i, y_{\text{ID}}^i) \sim \mathcal{P}_{X,Y}^{\text{ID}}, \text{ for } i \in \{1, \ldots, l\}\}$. Furthermore, $\mathcal{N}_l^y$ is a subset of $\mathcal{N}_l$ with ID data that below to the label of

$y$, namely, $\mathcal{N}_l^y = \{\mathbf{x}_{\text{ID}}^i | (\mathbf{x}_{\text{ID}}^i, y_{\text{ID}}^i) \sim \mathcal{P}_{X,Y}^{\text{ID}}$, and $y_{\text{ID}}^i = \hat{y}_{\text{ID}}$, for $i \in \{1, \ldots, l\}\}$. Accordingly, based on Theorem 1, we can ensure that learning from the auxiliary OOD detection task, i.e., small Eq. (5), can benefit real OOD detection, freeing from the mistaken OOD generation.

## 4.3 Overall Algorithm

We further summarize the training and the inferring procedure for the predictor of ATOL, given the crafted auxiliary ID and the auxiliary OOD distributions that satisfy **C**1 via Eqs. (6)-(7). The pseudo codes are further summarized in Appendix D due to the space limit.

**Training Procedure.** The overall learning objective consists of the real task learning, the auxiliary task learning, and the ID distribution alignment, which is given by:

$$
\min_{\mathbf{h} = \rho \circ \phi} \overbrace{\mathbb{E}_{\mathcal{P}_{X,Y}^{\text{ID}}} [\ell_{\text{CE}}(\mathbf{x}_{\text{ID}}, y_{\text{ID}}; \mathbf{h})]}^{\text{(a) real task learning}} + \overbrace{\mathbb{E}_{\mathcal{G}_{X,Y}^{\text{ID}}} [\ell_{\text{CE}}(\hat{\mathbf{x}}_{\text{ID}}, \hat{y}_{\text{ID}}; \mathbf{h})] + \lambda \mathbb{E}_{\mathcal{G}_X^{\text{OOD}}} [\ell_{\text{OE}}(\hat{\mathbf{x}}_{\text{OOD}}; \mathbf{h})]}^{\text{(b) auxiliary task learning}} +
$$
$$
\underbrace{\alpha \mathbb{E}_{\mathcal{G}_{X,Y}^{\text{ID}}} [\ell_{\text{align}}(\hat{\mathbf{x}}_{\text{ID}}, \hat{y}_{\text{ID}}; \phi)]}_{\text{(c) ID distribution alignment}},
\tag{10}
$$

where $\alpha, \lambda \geq 0$ are the trade-off parameters. By finding the predictor that leads to the minimum of the Eq. (10), our ATOL can ensure the improved performance of real OOD detection. Note that, Eq. (10) can be realized in a stochastic manner, suitable for deep model training.

**Inferring Procedure.** We adopt the MaxLogit scoring [17] in OOD detection. Given a test input $\mathbf{x}$, the MaxLogit score is given by:

$$
s_{\text{ML}}(\mathbf{x}; \mathbf{h}) = \max_k \mathbf{h}_k(\mathbf{x}),
\tag{11}
$$

where $\mathbf{h}_k(\cdot)$ denotes the $k$-th logit output. In general, the MaxLogit scoring is better than other commonly-used scoring functions, such as maximum softmax prediction [15], when facing large semantic space. Therefore, we choose MaxLogit instead of MSP for OOD scoring in our realization.

## 5 Experiments

This section conducts extensive experiments for ATOL in OOD detection. In Section 5.1, we describe the experiment setup. In Section 5.2, we demonstrate the main results of our method against the data generation-based counterparts on both the CIFAR [30] and the ImageNet [8] benchmarks. In Section 5.3, we further conduct ablation studies to comprehensively analyze our method.

### 5.1 Setup

**Backbones setups.** For the CIFAR benchmarks, we employ the WRN-40-2 [78] as the backbone model. Following [36], models have been trained for 200 epochs via empirical risk minimization, with a batch size 64, momentum 0.9, and initial learning rate 0.1. The learning rate is divided by 10 after 100 and 150 epochs. For the ImageNet, we employ pre-trained ResNet-50 [13] on ImageNet, downloaded from the PyTorch official repository.

**Generators setups.** For the CIFAR benchmarks, following [31], we adopt the generator from the Deep Convolutional (DC) GAN [51]). Following the vanilla generation objective, DCGAN has been trained on the ID data, where the batch size is 64 and the initial learning rate is 0.0002. For the ImageNet benchmark, we adopt the generator from the BigGAN [2] model, designed for scaling generation of high-resolution images, where the pre-trained model biggan-256 can be downloaded from the TensorFlow hub. Note that our method can work on various generators except those mentioned above, even with a random-parameterized one, later shown in Section 5.3.

**Baseline methods.** We compare our ATOL with advanced data generation-based methods in OOD detection, including (1) BoundaryGAN [31], (2) ConfGAN [55], (3) ManifoldGAN [62], (4) G2D [50] (5) CMG [67]. For a fair comparison, all the methods use the same generator and pre-trained backbone without regularizing with outlier data.

**Auxiliary task setups.** Hyper-parameters are chosen based on the OOD detection performance on validation datasets. In latent space, the auxiliary ID distribution is the high density region of the

Table 1: Comparison in OOD detection on the CIFAR and ImageNet benchmarks. $\downarrow$ (or $\uparrow$) indicates smaller (or larger) values are preferred; a bold font indicates the best results in a column.

| Methods | CIFAR-10 | | CIFAR-100 | | ImageNet | |
|---|---|---|---|---|---|---|
| | FPR95 $\downarrow$ | AUROC $\uparrow$ | FPR95 $\downarrow$ | AUROC $\uparrow$ | FPR95 $\downarrow$ | AUROC $\uparrow$ |
| BoundaryGAN [31] | 55.60 | 86.46 | 76.72 | 75.79 | 85.48 | 66.78 |
| ConfGAN [55] | 31.57 | 93.01 | 74.86 | 77.67 | 74.88 | 77.03 |
| ManifoldGAN [62] | 26.68 | 94.09 | 73.54 | 77.40 | 72.50 | 77.73 |
| G2D [50] | 31.83 | 91.74 | 70.73 | 79.03 | 74.93 | 77.16 |
| CMG [67] | 39.83 | 92.83 | 79.60 | 77.51 | 72.95 | 77.63 |
| **ATOL** (ours) | **14.66** | **97.05** | **55.22** | **87.24** | **51.80** | **85.82** |

**Mixture of Gaussian (MoG).** Each sub-Gaussian in the MoG has the mean $\boldsymbol{\mu} = (\mu_1, \mu_2, \ldots, \mu_m)$ and the same covariance matrix $\sigma \cdot \boldsymbol{I}$, where $m$ is the dimension of latent space, $\mu_i$ is randomly selected from the set $\{-\mu, \mu\}$, and $\boldsymbol{I}$ is the identity matrix. The auxiliary OOD distribution is the uniform distribution except for high MoG density region, where each dimension has the same space size $u$. For the CIFAR benchmarks, the value of $\alpha$ is set to 1, $\mu$ is 5, $\sigma$ is 0.1, and $u$ is 8. For the ImageNet benchmarks, the value of $\alpha$ is set to 1, $\mu$ is 5, $\sigma$ is 0.8, and $u$ is 8.

**Training details.** For the CIFAR benchmarks, ATOL is run for 10 epochs and uses SGD with an initial learning rate 0.01 and the cosine decay [37]. The batch size is 64 for real ID cases, 64 for auxiliary ID cases, and 256 for auxiliary OOD cases. For the ImageNet benchmarks, ATOL is run for 10 epochs using SGD with an initial learning rate 0.0003 and a momentum 0.9. The learning rate is decayed by a factor of 10 at 30%, 60%, and 90% of the training steps. Furthermore, the batch size is fixed to 32 for real ID cases, 32 for auxiliary ID cases, and 128 for auxiliary OOD cases.

**Evaluation metrics.** The OOD detection performance of a detection model is evaluated via two representative metrics, which are both threshold-independent [7]: the false positive rate of OOD data when the true positive rate of ID data is at 95% (FPR95); and the *area under the receiver operating characteristic curve* (AUROC), which can be viewed as the probability of the ID case having greater score than that of the OOD case.

## 5.2 Main Results

We begin with our main experiments on the CIFAR and ImageNet benchmarks. Model performance is tested on commonly-used OOD datasets. For the CIFAR cases, we employed Texture [6], SVHN [47], Places365 [84], LSUN-Crop [76], LSUN-Resize [76], and iSUN [74]. For the ImageNet case, we employed iNaturalist [18], SUN [74], Places365 [84], and Texture [6]. In Table 1, we report the average performance (i.e., FPR95 and AUROC) regarding the OOD datasets mentioned above. Please refer to Tables 17-18 and 19 in Appendix F.1 for the detailed reports.

**CIFAR benchmarks.** Overall, our ATOL can lead to effective OOD detection, which generally demonstrates better results than the rivals by a large margin. In particular, compared with the advanced data generation-based method, e.g., ManifoldGAN [62], ATOL significantly improves the performance in OOD detection, revealing 12.02% and 2.96% average improvements w.r.t. FPR95 and AUROC on the CIFAR-10 dataset and 18.32% and 9.84% of the average improvements on the CIFAR-100 dataset. This highlights the superiority of our novel data generation strategy, relieving mistaken OOD generation as in previous methods.

**ImageNet benchmark.** We evaluate a large-scale OOD detection task based on ImageNet dataset. Compared to the CIFAR benchmarks above, the ImageNet task is more challenging due to the large semantic space and large amount of training data. As we can see, previous data generation-based methods reveal poor performance due to the increased difficulty in searching for the proper OOD-like data in ImageNet, indicating the critical mistaken OOD generation remains (cf., Appendix F.9). In contrast, our ATOL can alleviate this issue even for large-scale datasets, e.g., ImageNet, thus leading to superior results. In particular, our ATOL outperforms the best baseline ManifoldGAN [62] by 20.70% and 8.09% average improvement w.r.t. FPR95 and AUROC, which demonstrates the advantages of our ATOL to relieve mistaken OOD generation and benefit to real OOD detection.

Table 2: Effectiveness of auxiliary task crafting and learning scheme of ATOL. ↓ (or ↑) indicates smaller (or larger) values are preferred; a bold font indicates the best results in a row.

| Ablation | | ATOL w/o $\tau$ | ATOL w/o $\ell_{\text{reg}}$ | ATOL w/o $\ell_{\text{align}}$ | ATOL |
|---|---|---|---|---|---|
| CIFAR-10 | FPR95 ↓ | 87.64 | 41.13 | 22.29 | **14.54** |
|  | AUROC ↑ | 59.32 | 90.38 | 94.08 | **97.01** |
| CIFAR-100 | FPR95 ↓ | 92.90 | 82.80 | 74.20 | **55.06** |
|  | AUROC ↑ | 51.34 | 73.95 | 80.56 | **87.26** |

Table 3: Performance comparisons with different setups of the generator on CIFAR benchmarks; ↓ (or ↑) indicates smaller (or larger) values are preferred; a bold font indicates the best results in a row.

| Generators | | DCGAN | Rand-DCGAN | StyleGAN | BigGAN |
|---|---|---|---|---|---|
| CIFAR-10 | FPR95 ↓ | 14.66 | 20.78 | 13.65 | **8.61** |
|  | AUROC ↑ | 97.05 | 95.57 | 96.63 | **97.90** |
| CIFAR-100 | FPR95 ↓ | 55.52 | 65.69 | 42.62 | **36.72** |
|  | AUROC ↑ | 86.21 | 83.26 | 89.17 | **91.33** |

## 5.3 Ablation Study

To further demonstrate the effectiveness of our ATOL, we conduct extensive ablation studies to verify the contributions of each proposed component.

**Auxiliary task crafting schemes.** In Section 4.1, we introduce the realization of the auxiliary OOD detection task in a tractable way. Here, we study two components of generation: the disjoint supports for ID and OOD distributions in the latent space and the distance-preserving generator. In particular, we compare ATOL with two variants: 1) ATOL $w/o\ \tau$, where the ID and OOD distributions may overlap in latent space without separation via the threshold $\tau$; 2) ATOL $w/o\ \ell_{\text{reg}}$, where we directly use the generator to generate the auxiliary data without further regularization. As we can see in Table 2, the overlap between the auxiliary ID and the auxiliary OOD distributions in latent space will lead to the failure of the auxiliary OOD detection task, demonstrating the catastrophic performance (e.g., $92.20\%$ FPR95 and $51.34\%$ AUROC on CIFAR-100). Moreover, without the regularized generator as a distance-preserving function, the ambiguity of generated data will compromise the performance of ATOL, which verifies the effectiveness of our generation scheme.

**Auxiliary task learning schemes.** In Section 4.2, we propose ID distribution alignment to transfer model capability from auxiliary OOD detection task to real OOD detection. In Table 2, we conduct experiments on a variant, namely, ATOL $w/o\ \ell_{\text{align}}$, where the predictor directly learns from the auxiliary task without further alignment. Accordingly, ATOL $w/o\ \ell_{\text{align}}$ can reveal better results than ATOL $w/o\ \ell_{\text{reg}}$, with $18.84\%$ and $3.70\%$ further improvement w.r.t. FPR95 and AUROC on CIFAR-10 and $8.60\%$ and $6.61\%$ on CIFAR-100. However, the predictor learned on the auxiliary task cannot fit into the real OOD detection task. Thus, we further align the distributions between the auxiliary and real ID data in the transformed space, significantly improving the OOD detection performance with $7.75\%$ on CIFAR-10 and $19.14\%$ on CIFAR-100 w.r.t. FPR95.

**Generator Setups.** On CIFAR benchmarks, we use the generator of the DCGAN model fitting on the ID data in our ATOL. We further investigate whether ATOL remains effective when using generators with other setups, summarized in Table 3. First, we find that even with a random-parameterized DCGAN generator, i.e., Rand-DCGAN, our ATOL can still yield reliable performance surpassing its advanced counterparts. Such an observation suggests that our method requires relatively low training costs, a property not shared by previous methods. Second, we adopt more advanced generators, i.e., BigGAN [2] and StyleGAN-v2 [27]. As we can see, ATOL can benefit from better generators, leading to large improvement $18.80\%$ and $5.12\%$ w.r.t. FPR95 and AUROC on CIFAR-100 over the DCGAN case. Furthermore, even with complicated generators, our method demonstrates promising performance improvement over other methods with acceptable computational resources. Please refer to the Appendix E.8 and Appendix F.7 for more details.

**Quantitative analysis on computation.** Data generation-based methods are relatively expensive in computation, so the consumed training time is also of our interest. In Table 4, we compare the time costs of training for the considered data generation-based methods. Since our method does not need a complex generator training procedure during the OOD learning step, we can largely alleviate the expensive cost of computation inherited from the data generation-based approaches. Overall, our method requires minimal training costs compared to the advanced counterparts, which further reveals the efficiency of our method. Therefore, our method can not only lead to improved performance over previous methods, but it also requires fewer computation resources.

Table 4: Computational comparison with other counterparts. We report the per-epoch training time (measured by seconds) on CIFAR benchmarks.

| Methods | Training time (s) |
|---|---|
| BoundaryGAN | $106.98 \pm 7.82$ |
| ConfGAN | $111.37 \pm 5.72$ |
| ManifoldGAN | $159.23 \pm 9.23$ |
| G2D | $73.29 \pm 4.09$ |
| CMG | $254.22 \pm 2.42$ |
| ATOL | $\mathbf{57.22} \pm 2.28$ |

## 6   Conclusion

Data generation-based methods in OOD detection are a promising way to make the predictor learn to discern ID and OOD data without real OOD data. However, mistaken OOD generation significantly challenges their performance. To this end, we propose a powerful data generation-based learning method by the proposed auxiliary OOD detection task, largely relieving mistaken OOD generation. Extensive experiments show that our method can notably improve detection performance compared to advanced counterparts. Overall, our method can benefit applications for data generation-based OOD detection, and we intend further research to extend our method beyond image classification. We also hope our work can draw more attention from the community toward data generation-based methods, and we anticipate our auxiliary task-based learning scheme can benefit more directions in OOD detection, such as wild OOD detection [28].

## Acknowledgments and Disclosure of Funding

HTZ, QZW and BH were supported by the NSFC Young Scientists Fund No. 62006202, NSFC General Program No. 62376235, Guangdong Basic and Applied Basic Research Foundation No. 2022A1515011652, HKBU Faculty Niche Research Areas No. RC-FNRA-IG/22-23/SCI/04, and HKBU CSD Departmental Incentive Scheme. XBX and TLL were partially supported by the following Australian Research Council projects: FT220100318, DP220102121, LP220100527, LP220200949, and IC190100031. FL was supported by Australian Research Council (ARC) under Award No. DP230101540, and by NSF and CSIRO Responsible AI Program under Award No. 2303037.

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
