# A  Notations

In this section, we summarize the adopted notations in Table 5.

Table 5: Main notations and their descriptions.

| Notation | Description |
|---|---|
| | Spaces |
| $\mathcal{X}$ and $\mathcal{Y}$ | the data space and the label space $\{1, \cdots, c\}$ |
| $n$ and $c$ | the dimension of the data space and the dimension of the label space |
| $d$ and $m$ | the dimension of the embedding space and the dimension of the latent space |
| | Distributions and Sets |
| $\mathcal{P}_{X,Y}^{ID}$ and $\mathcal{P}_X^{ID}$ | the joint and the marginal real ID distribution |
| $\mathcal{P}_X^{OOD}$ | the marginal real OOD distribution |
| $\mathcal{G}_{X,Y}^{ID}$ and $\mathcal{G}_X^{ID}$ | the joint and the marginal auxiliary ID distribution |
| $\mathcal{G}_X^{OOD}$ | the marginal auxiliary OOD distribution |
| $\mathcal{M}_Z$ | the specified MoG distribution |
| $\mathcal{U}_Z$ | the specified uniform distribution |
| | Data |
| $\mathbf{x}_{ID}$ and $y_{ID}$ | the real ID data and label |
| $\hat{\mathbf{x}}_{ID}$ and $\hat{y}_{ID}$ | the auxiliary ID data and label |
| $\hat{\mathbf{x}}_{OOD}$ | the auxiliary OOD data |
| $\mathbf{z}_{ID}$ and $\mathbf{z}_{OOD}$ | the latent ID and the latent OOD data |
| $\mathcal{Z}^{ID}$ and $\mathcal{Z}^{OOD}$ | the latent ID and the latent OOD data sets |
| | Models |
| $\mathbf{h}$ | the predictor: $\mathbb{R}^n \to \mathbb{R}^c$ |
| $\phi$ and $\rho$ | the feature extractor and the classifier |
| $s(\cdot; \mathbf{h})$ | the scoring function: $\mathbb{R}^n \to \mathbb{R}$ |
| $f_\beta(\cdot)$ | the OOD detector: $\mathcal{X} \to \{ID, OOD\}$, with threshold $\beta$ |
| $G$ | the generator: $\mathbb{R}^m \to \mathbb{R}^n$ |
| | Loss and Function |
| $\ell_{CE}$ and $\ell_{OE}$ | ID loss function and OOD loss function |
| $\ell_{reg}$ | the generator regularization loss function |
| $\ell_{align}$ | the alignment loss |
| $\phi'(\cdot)$ | the general mapping function |
| $\mathcal{M}(\cdot)$ | the density function of MoG |

# B  Related Works

**OOD Scoring Functions**. To discern OOD data from ID data, many works study how to devise effective *OOD scoring functions*, typically assuming a well-trained model on ID data with its fixed parameters [15, 36]. As a baseline method, Hendrycks and Gimpel [15] takes the *maximum softmax prediction* (MSP) as the OOD score, where, in expectation, the MSP should be low for OOD data since their true labels are not in the considered label space. However, MSP frequently makes mistakes due to the over-confidence issue [36]. Therefore, recent works devise improved scoring strategies [36, 20, 17, 64, 57]; integrate gradient information [25, 21, 35] or embedding features [32, 53, 9, 46]; make further adjustments on specific tasks [56, 68, 38, 39]. In Appendix F.6, we test ATOL with different scoring strategies, demonstrating that proper scoring functions can lead to improved performance for data generation-based OOD detection. Therefore, OOD scoring functions are typically orthogonal to data generation-based approaches.

**OOD Training Strategies**. OOD detection can also be improved by model fine-tuning, motivating advanced works studying OOD training strategies. As one of the most potent approaches, *outlier exposure* [16] train to make the perdictor produce low-confidence predictions for OOD data. Based on outlier exposure, a set of improved methods have been proposed, from the perspective of data re-sampling [79, 42], background classification [34, 3, 45], data transformation [70], adversarial robust learning [34, 14, 4], meta learning [26] and energy scoring [36, 28]. However, real OOD data are hard to be accessed, largely hindering their practical validity.

Therefore, a set of learning strategies has been proposed, considering situations where real OOD data are unavailable. Therein, improved representation [58, 54, 65, 77, 71, 44], extra-class learning [20, 52], and pseudo features learning [10, 60] have demonstrated their strengths for improved detection. However, these methods can hardly beat the outlier exposure-based methods. Therefore, several works [31, 62, 55, 50, 67] adopt data generators to synthesize OOD data for model training. They can make the predictor learn to discern ID and OOD patterns without tedious OOD data acquisition. However, the data generator may wrongly generate unreliable data containing ID semantics, confusing the predictor between ID and OOD cases.

## C  Proof of Proposition 1

*Proof.* Following Theorem 3 in Fang et al. [11], if Eq. (5) approaches 0 and **C**1 holds, the predictor $\mathbf{h} = \boldsymbol{\rho} \circ \boldsymbol{\phi}$ can separate the auxiliary ID and the auxiliary OOD cases well, i.e., $\mathrm{supp}\phi_{\#}\mathcal{G}_X^{\mathrm{ID}} \cap \mathrm{supp}\phi_{\#}\mathcal{G}_X^{\mathrm{OOD}} = \emptyset$. Then, by aligning the real and the auxiliary ID distributions in transformed space (cf. **C**2), we can conclude that the auxiliary OOD data are almost reliable w.r.t. the real ID data. $\quad\square$

## D  Overall Algorithm

In this section, we discuss our learning framework in detail. Our ATOL consists of two stages: 1) generator regularization and 2) auxiliary task OOD learning.

For the generator regularization, the overall training framework is summarized in Algorithm 1, regularizing in a stochastic manner with `num_step_g` iterations. We have the generator $G : \mathbb{R}^m \to \mathbb{R}^n$ and the pre-defined latent distribution $\mathcal{U}_Z$ in the latent space. In each training step, we sample a set of latent data from $\mathcal{U}_Z$, assuming be of the size $b$ as that of the mini-batch. With the regularization term, we update the generator via one step of mini-batch gradient descent.

---

**Algorithm 1** Generator Regularization.

---

1: **Inputs:** initialized generator $G(\cdot)$ and the pre-defined latent distribution $\mathcal{U}_Z$.
2: **for** $t = 1$ **to** `num_step_g` **do**
3:     Sample latent data $\{\mathbf{z}\}_{i=1}^{b}$ from $\mathcal{U}_Z$;
4:     Compute regularization risk $\ell_{\mathrm{reg}}(\mathbf{z}; G)$;
5:     Update generator $\min_G \ell_{\mathrm{reg}}(\mathbf{z}; G)$;
6: **end for**
7: **Output:** regularized generator $G(\cdot)$.

---

For the auxiliary task OOD learning, the overall training framework is summarized in Algorithm 2, optimizing the predictor in a stochastic manner with `num_step_p` iterations. In each training step, with the regularized generator, the auxiliary ID data $\hat{\mathbf{x}}_{\mathrm{ID}}$ and the auxiliary OOD data $\hat{\mathbf{x}}_{\mathrm{OOD}}$ can be generated from the latent ID data $\mathbf{z}_{\mathrm{ID}}$ and the latent OOD data $\mathbf{z}_{\mathrm{OOD}}$ respectively, where the latent ID and OOD data follow the crafted distribution $\mathcal{Z}_{\mathrm{ID}}, \mathcal{Z}_{\mathrm{OOD}}$[3] in Eqs (6)-(7). We assume the size $b$ as that of the mini-batch regarding the ID samples and $b'$ for the OOD samples[4]. Then, the risk for the auxiliary and the real data are jointly computed and update the predictor $\mathbf{h}$ parameters using Eq. (10). After training, we apply the MaxLogit scoring in discerning ID and OOD cases.

## E  Details of Experiment Configuration

### E.1  Hyper-parameters

We study the effect of hyper-parameters on the final performance of our ATOL, where we consider the trade-off parameter $\alpha$, mean value $\mu$, covariance matrix scale $\sigma$ for *Mixture of Gaussian*(MoG), the sampling space size $u$ in latent space, and the dimension of latent space $m$. We also study the impact

---

[3]With abuse of notation, we denote the distribution in latent space as $\mathcal{Z}$ for simplicity.

[4]Note that the mini-batch size of the real ID data and the auxiliary ID data have no need to be equal in the Algorithm 2. In this paper we make them equal for convenience

---

**Algorithm 2** Auxiliary Task OOD Learning.

---

1: **Inputs:** predictor $\mathbf{h} = \boldsymbol{\rho} \circ \boldsymbol{\phi}$, regularized generator $G(\cdot)$, real ID distribution $\mathcal{P}^{\mathrm{ID}}_{X,Y}$, crafted latent distributions $\mathcal{Z}_{\mathrm{ID}}$ and $\mathcal{Z}_{\mathrm{OOD}}$;
2: **for** $t = 1$ **to** `num_step_p` **do**
3:     Sample latent data $\{\mathbf{z}_{\mathrm{ID}}\}_{i=1}^{b}$ and $\{\mathbf{z}_{\mathrm{OOD}}\}_{i=1}^{b'}$ from $\mathcal{Z}_{\mathrm{ID}}$ and $\mathcal{Z}_{\mathrm{OOD}}$, resp;
4:     Generate auxiliary data $\hat{\mathbf{x}}_{\mathrm{ID}} = G(\mathbf{z}_{\mathrm{ID}})$ and $\hat{\mathbf{x}}_{\mathrm{OOD}} = G(\mathbf{z}_{\mathrm{OOD}})$ by regularized generator $G(\cdot)$ and sample real ID data $\{(\mathbf{x}_{\mathrm{ID}}, y_{\mathrm{ID}})\}_{i=1}^{b}$ from $\mathcal{P}^{\mathrm{ID}}_{X,Y}$;
5:     Compute risk $\ell(\mathbf{h}) = \ell_{\mathrm{CE}}(\mathbf{x}_{\mathrm{ID}}, y_{\mathrm{ID}}; \hat{\mathbf{x}}_{\mathrm{ID}}, \hat{y}_{\mathrm{ID}}; \mathbf{h}) + \ell_{\mathrm{OE}}(\hat{\mathbf{x}}_{\mathrm{OOD}}; \mathbf{h}) + \ell_{\mathrm{align}}(\hat{\mathbf{x}}_{\mathrm{ID}}, \hat{y}_{\mathrm{ID}}; \boldsymbol{\phi})$;
6:     Update predictor $\min_{\mathbf{h}} \ell(\mathbf{h})$;
7: **end for**
8: **Output:** predictor $\mathbf{h}(\cdot)$.

---

of transformed data from different layers of WRN-40-2. All the above experiments are conducted on the CIFAR benchmarks.

**Trade-off parameter $\alpha$.** Table 6 demonstrates the performance of ATOL with varying $\alpha$ values that trade-off the OOD detection task learning loss and the alignment losses. $\alpha$ is selected from $\{0.01, 0.1, 0.5, 1, 3, 5, 10\}$. We can observe that on CIFAR-10, the best performance is obtained at $\alpha = 1$, and the best performance on CIFAR-100 is obtained at $\alpha = 5$. In general, a large $\alpha$ is advantageous for the accuracy of the predictor since ID distribution alignment not only enables the transfer of the ability to discern ID and OOD data but also facilitates the ID classification of the predictor. However, when $\alpha$ is large, the predictor tends to align the ID data rather than discern ID and OOD cases at the early stage of training. Hence, a relatively small ($\alpha \leq 5$) usually leads to good performance than a much larger value.

Table 6: Performance of ATOL with varying $\alpha$ on CIFAR-10 and CIFAR-100

| Trade-off parameter $\alpha$ | CIFAR-10 | | | CIFAR-100 | | |
|---|---|---|---|---|---|---|
| | FPR95 $\downarrow$ | AUROC $\uparrow$ | Acc ID $\uparrow$ | FPR95 $\downarrow$ | AUROC $\uparrow$ | Acc ID $\uparrow$ |
| $\alpha = 0.01$ | 22.99 | 92.66 | 92.89 | 71.07 | 80.45 | 72.67 |
| $\alpha = 0.1$ | 20.02 | 94.66 | 93.43 | 65.99 | 82.92 | 73.44 |
| $\alpha = 0.5$ | 15.56 | 96.05 | 94.03 | 60.23 | 84.88 | 73.89 |
| $\alpha = 0.8$ | 15.51 | 96.40 | 94.15 | 60.23 | 84.93 | 73.99 |
| $\alpha = 1$ | **14.11** | **96.94** | 94.17 | 59.89 | 84.77 | 73.99 |
| $\alpha = 3$ | 16.01 | 96.56 | 93.96 | 56.53 | 86.37 | 73.89 |
| $\alpha = 5$ | 16.28 | 96.57 | 94.17 | **55.38** | **86.41** | 73.98 |
| $\alpha = 10$ | 17.08 | 96.50 | **94.57** | 61.82 | 84.97 | **74.58** |

**Mean values for MoG $\mu$.** We show the effect of the mean value $\mu$ of sub-Gaussian from the Mixture of Gaussian $\mathcal{M}$ in the latent space. The mean value $\mu$ decides the latent distribution for the auxiliary ID data, and the results are listed in Table 7. We conduct experiments with two realizations of the Gaussian mean: case 1 is to choose at random from a uniform distribution, and case 2 (used in ATOL) is to obtain the values from the vertices of a high-dimensional hypercube (i.e., a mean vector with $m$ elements, where each element is randomly selected from between $\{-\mu, \mu\}$ as the mean for MoG).

Specifically, the performance of case 1 is stable, since the randomness of value sampling. In contrast, in case 2, the mean value plays an important role in the performance of ATOL, where the mean for each sub-distribution is fixed at a set value. In general, a relatively large mean value for the MoG usually leads to good performance than a much smaller value. When the mean value is small, the MoG will concentrate on a limited region, leading to the great overlapping among sub-Gaussians. Such an overlapping can result in confusion of the semantics of the auxiliary ID data. Moreover, a proper choice value for the case 2 is superior to the case 1, reflecting that our strategy is useful for our proposed ATOL.

**Covariance matrix for MoG $\sigma$.** We show the effect of $\sigma$ for the Mixture of Gaussian $\mathcal{M}$ in the latent space. The value $\sigma$ decides the covariance matrix of the latent distribution for the auxiliary ID data. We vary $\sigma \in \{0.05, 0.1, 0.3, 0.5, 1, 1.5\}$, and the results are listed in Table 8. As we can see, too large $\sigma$ leads to inferior performance since too large $\sigma$ will result in largely overlapping with other Gaussian in latent space, as we discussed earlier. On CIFAR-10 dataset, the best performance is

Table 7: Performance of ATOL with varying $\mu$ on CIFAR-10 and CIFAR-100

| Mean $\mu$ | | Case 1 | | | | | | | Case 2 | | | | | | |
|---|---|---|---|---|---|---|---|---|---|---|---|---|---|---|---|
| | | 0.1 | 0.5 | 1 | 1.5 | 2 | 3 | 5 | 0.1 | 0.5 | 1 | 1.5 | 2 | 3 | 5 |
| CIFAR-10 | FPR95 ↓ | 17.60 | 16.26 | 16.26 | 16.20 | 16.30 | 16.29 | 16.31 | 23.29 | 21.32 | 18.62 | 16.08 | 15.14 | 14.69 | **14.52** |
| | AUROC ↑ | 96.13 | 96.59 | 96.56 | 96.55 | 96.56 | 96.55 | 96.54 | 93.44 | 94.61 | 95.85 | 96.37 | 96.75 | 97.02 | **97.05** |
| CIFAR-100 | FPR95 ↓ | 63.93 | 64.04 | 63.42 | 62.55 | 63.05 | 63.84 | 63.56 | 62.08 | 63.17 | **55.24** | 59.49 | 60.06 | 61.80 | 62.54 |
| | AUROC ↑ | 83.26 | 83.37 | 83.54 | 84.42 | 84.14 | 83.80 | 84.08 | 84.19 | 82.28 | 86.35 | **86.98** | 85.72 | 85.22 | 84.85 |

obtained at $\sigma = 0.1$, while on CIFAR-100, the best performance is obtained at $\sigma = 0.5$. We suppose that the semantics and scale of the varying datasets differ, necessitating different $\sigma$.

Table 8: Performance of ATOL with varying $\sigma$ on CIFAR-10 and CIFAR-100

| Covariance matrix scale $\sigma$ | | $\sigma = 0.01$ | $\sigma = 0.1$ | $\sigma = 0.3$ | $\sigma = 0.5$ | $\sigma = 1$ | $\sigma = 1.5$ |
|---|---|---|---|---|---|---|---|
| CIFAR-10 | FPR95 ↓ | 14.67 | **14.62** | 14.88 | 15.99 | 20.09 | 21.31 |
| | AUROC ↑ | 96.99 | **97.02** | 96.99 | 96.37 | 95.45 | 94.61 |
| CIFAR-100 | FPR95 ↓ | 63.97 | 62.82 | 60.44 | **56.06** | 63.72 | 64.42 |
| | AUROC ↑ | 84.50 | 84.85 | 85.70 | **86.49** | 82.95 | 83.58 |

**Space size for latent space** $u$. We show the effect of $u$ for the latent distribution $\mathcal{U}_Z$, which identifies the space size of $\mathcal{U}_Z$ in latent space for the auxiliary OOD data. We vary $u \in \{0.5, 1, 4, 8, 16, 32\}$, and the results are listed in Table 9. Generally, a small latent space leads to the limited information in the latent space, which results in the inferior performance (e.g., the ATOL performance when $u = 0.5$). As we set a relatively large value ($u \geq 4$), the performances are stable on both datasets.

Table 9: Performance of ATOL with varying $u$ on CIFAR-10 and CIFAR-100

| Latent space size $u$ | | $u = 0.5$ | $u = 1$ | $u = 4$ | $u = 8$ | $u = 16$ | $u = 32$ |
|---|---|---|---|---|---|---|---|
| CIFAR-10 | FPR95 ↓ | 18.15 | 14.90 | 13.96 | 13.90 | 14.12 | 14.02 |
| | AUROC ↑ | 93.96 | 95.97 | 96.96 | 96.96 | 96.95 | 96.96 |
| CIFAR-100 | FPR95 ↓ | 60.18 | 58.02 | 56.25 | 56.42 | 56.09 | 55.97 |
| | AUROC ↑ | 83.42 | 85.43 | 86.64 | 86.57 | 86.38 | 86.44 |

**Dimension of latent space** $m$. To study the impact of dimension $m$ of latent space (the input space of the data generator), we conduct experiments based on different dimensions of a random-parameterized generator of DCGAN. We vary $m \in \{8, 16, 32, 64, 128, 256\}$, and the results are listed in Table 10. We find that the generator with $m = 64$ has the best performance. We suppose that a data generator with high dimensional input space may be more intractable, while low dimensional input space may not contain sufficient information. Therefore, a reasonably large $m$ helps achieve a better result.

The above experiments about hyper-parameters only serve to support the validity of our method. However, a proper choice of hyper-parameters can truly induce improved results in effective OOD detection, reflecting that all the introduced hyper-parameters are useful in our proposed ATOL.

## E.2 Effect of Mistaken OOD Generation

In section 1, we revisit the common baseline approach [16], which uses real OOD data, for OOD detection. We investigate the effect of mistaken OOD generation on the OOD detection performance. In particular, we use a WRN-40-2 architecture trained on CIFAR-100 with varying proportions of real ID data mixed in the real OOD data, reflecting the severity of wrong OOD data during training. As shown in Figure 1(b), the performance (FPR95) degrades rapidly from 35.70% to 64.21% as the proportion of unreliable OOD data increases from 0% to 75%. This trend signifies that current OOD detection methods are indeed challenged by mistaken OOD generation, which motivates our work.

Table 10: Performance of ATOL with varying $m$ on CIFAR-10 and CIFAR-100

| Dimension $m$ | | $m = 8$ | $m = 16$ | $m = 32$ | $m = 64$ | $m = 128$ | $m = 256$ |
|---|---|---|---|---|---|---|---|
| CIFAR-10 | FPR95 ↓ | 24.98 | 22.71 | 22.62 | 20.78 | 28.39 | 29.55 |
| | AUROC ↑ | 94.74 | 94.84 | 94.81 | 95.57 | 94.11 | 93.12 |
| CIFAR-100 | FPR95 ↓ | 70.62 | 70.03 | 68.33 | 65.69 | 65.93 | 71.83 |
| | AUROC ↑ | 80.45 | 80.65 | 82.30 | 83.26 | 82.86 | 78.90 |

### E.3 Realizations of Distance

As described in 4.1, we suppose that the centralized distance between two data points that are measured in the latent space should be positively correlated to that of the distance measured in the data space, namely, distance-preserving. In this ablation, we contrast the performance of different realizations used for the normalized distance in Eq. 8. We empirically test three realizations: 1) *Cosine similarity*-based: $d(\mathbf{z}_1, \mathbf{z}_2) = \frac{\mathbf{z}_1^\top \mathbf{z}_2}{\|\mathbf{z}_1\|\|\mathbf{z}_2\|}$; 2) *Taxicab distance*-based: $d(\mathbf{z}_1, \mathbf{z}_2) = \|\mathbf{z}_1 - \mathbf{z}_2\|_1 - \mathbb{E}\|\mathbf{z}_1 - \mathbf{z}_2\|_1$; 3) *Euclidean distance*-based [73]: $d(\mathbf{z}_1, \mathbf{z}_2) = \|\mathbf{z}_1 - \mathbf{z}_2\|_2 - \mathbb{E}\|\mathbf{z}_1 - \mathbf{z}_2\|_2$. As one can see from the Table 11, all the forms of distance can lead to reliable OOD detection performance, indicating that ATOL is general to various realizations.

Table 11: Performance comparisons with different realizations of centralized distance

| Centralized distance $d$ | CIFAR-10 | | CIFAR-100 | |
|---|---|---|---|---|
| | FPR95 ↓ | AUROC ↑ | FPR95 ↓ | AUROC ↑ |
| Cosine similarity-based | 19.73 | 95.77 | 60.15 | 85.41 |
| Taxicab distance-based | 18.40 | 94.78 | 58.76 | 85.83 |
| Euclidean distance-based | **14.62** | **97.02** | **55.52** | **86.32** |

### E.4 Class-agnostic Auxiliary ID Data

ATOL adopts a class-conditional way to generate auxiliary ID data [59, 72], then aligns transformed distribution with real ID data based on labels. In this ablation, we contrast with a class-agnostic implementation, i.e., we generate the auxiliary ID data from one single Gaussian and randomly assign labels to the auxiliary ID data. The parameter of the single Gaussian is similar to the sub-Gaussian in the MoG, except that the mean value is the zero vector. Under the same training setting, the class-agnostic way for auxiliary ID data leads to a worse result, which may be caused by random labels and unavailable alignment. Even if class-agnostic approaches struggle due to the homogenization of auxiliary data, the predictor can still learn to discern ID and OOD data.

Table 12: Performance comparisons with different implementations of auxiliary ID data

| Methods | CIFAR-10 | | CIFAR-100 | |
|---|---|---|---|---|
| | FPR95 ↓ | AUROC ↑ | FPR95 ↓ | AUROC ↑ |
| Class-agnostic | 21.72 | 93.93 | 61.54 | 84.34 |
| Class-conditional | **14.62** | **97.02** | **55.52** | **86.32** |

### E.5 Sample More Auxiliary Data

We note that with a powerful generation-based learning scheme, we can generate sufficient data to make the predictor learn the knowledge of OOD without tedious OOD data acquisition. Therefore, we consider increasing the number of generated auxiliary data, making the predictor see more auxiliary data to strengthen the ability to discern ID and OOD data. To verify the effect of generating more data, we conduct ATOL on CIFAR-10 and CIFAR-100 with different batch size $b$ and $b'$ w.r.t. the ID and OOD data, respectively. The experiment results are shown in Table 13, demonstrating that generating more auxiliary data could strengthen the effect of ATOL. However, larger batch size means the extra calculation cost, which may be improved in the future.

Table 13: Performance of ATOL with varying batch size of ID and OOD data

| Batch size | | CIFAR-10 | | CIFAR-100 | |
|---|---|---|---|---|---|
| $b$ | $b'$ | FPR95 ↓ | AUROC ↑ | FPR95 ↓ | AUROC ↑ |
| $b = 32$ | $b' = 32$ | 22.21 | 95.04 | 65.94 | 83.67 |
| $b = 32$ | $b' = 128$ | 15.43 | 96.83 | 58.77 | 83.98 |
| $b = 64$ | $b' = 64$ | 19.60 | 95.38 | 63.26 | 84.12 |
| $b = 64$ | $b' = 256$ | 14.21 | 96.97 | 55.03 | 86.16 |
| $b = 128$ | $b' = 128$ | 20.39 | 95.43 | 63.62 | 82.33 |
| $b = 128$ | $b' = 512$ | 13.99 | 97.09 | 54.96 | 86.17 |
| $b = 256$ | $b' = 256$ | 19.25 | 95.72 | 62.94 | 83.97 |
| $b = 256$ | $b' = 1024$ | **13.68** | **97.18** | **54.71** | **86.22** |

## E.6 Using Embedding of Different Layers

To study the impact of embedding spaces from different layers, we conduct ID distribution alignment on different output layers of WRN-40-2. In Table 14, we find that using the embedding space from the penultimate layer of WRN-40-2 achieves the best performance. We suppose that alignment based on a too-shallow layer may not be enough to impact the embedding of subsequent layers. However, the calibration based on the last layer may interrupt the normal classification between the real data and auxiliary data, since that they have completely different high-level semantics.

Table 14: Performance comparisons with different layers on CIFAR-10 and CIFAR-100

| Different embedding layers | CIFAR-10 | | CIFAR-100 | |
|---|---|---|---|---|
| | FPR95 ↓ | AUROC ↑ | FPR95 ↓ | AUROC ↑ |
| Block 1 ($h_{\text{shallow}}$) | 31.78 | 91.01 | 72.88 | 80.15 |
| Block 2 ($h_{\text{middle}}$) | 23.43 | 93.62 | 70.04 | 81.52 |
| Block 3 ($h_{\text{deep}}$) | **13.68** | **97.18** | **55.52** | **86.32** |
| Last layer ($h_{\text{last}}$) | 27.11 | 91.85 | 65.12 | 83.28 |

## E.7 Using Different Network Architectures

In the main paper, we have shown that ATOL is competitive on WRN-40-2. The following experimental results on CIFAR benchmarks can support our claims 1, where using a more complex model (i.e., DenseNet-121 [19]) can lead to better performance in OOD detection. All the numbers are reported over OOD test datasets described in Section 5.1.

Table 15: Performance comparisons with different network architectures DenseNet-121 on CIFAR-10 and CIFAR-100 ↓ (or ↑) indicates smaller (or larger) values are preferred.

| Method | SVHN | | LSUN-Crop | | LSUN-Resize | | iSUN | | Texture | | Places365 | | Average | |
|---|---|---|---|---|---|---|---|---|---|---|---|---|---|---|
| | FPR95 ↓ | AUROC ↑ | FPR95 ↓ | AUROC ↑ | FPR95 ↓ | AUROC ↑ | FPR95 ↓ | AUROC ↑ | FPR95 ↓ | AUROC ↑ | FPR95 ↓ | AUROC ↑ | FPR95 ↓ | AUROC ↑ |
| CIFAR-10 | | | | | | | | | | | | | | |
| WRN-40-2 | 20.60 | 96.03 | 1.48 | 99.59 | 5.20 | 98.78 | 5.00 | 98.76 | 26.05 | 95.03 | 27.55 | 94.33 | 14.31 | 97.09 |
| DenseNet-121 | 18.05 | 96.27 | 1.45 | 99.58 | 4.35 | 99.88 | 4.45 | 98.86 | 20.90 | 95.70 | 25.85 | 94.39 | 12.51 | 97.28 |
| CIFAR-100 | | | | | | | | | | | | | | |
| WRN-40-2 | 70.85 | 84.70 | 13.45 | 97.52 | 51.85 | 90.12 | 55.80 | 89.02 | 63.10 | 83.37 | 75.30 | 78.86 | 55.06 | 87.26 |
| DenseNet-121 | 70.30 | 85.96 | 61.55 | 87.93 | 22.15 | 95.87 | 22.30 | 95.59 | 48.30 | 87.87 | 77.15 | 79.28 | 50.29 | 88.75 |

## E.8 How to best perform data generation-based methods for OOD detection

BoundaryGAN [31], ConfGAN [55] and ManifoldGAN [62] use DCGAN [51] to generate OOD data to benefit the predictor for OOD detection. As we show in Section 5.2, even with the DCGAN, ATOL has shown promising OOD detection performance. Moreover, we argue for ATOL can profit from a delicate data generators [5], which can generate more diverse data to further benefit the predictor learning from generated data. To this end, we use the generator of StyleGAN-v2 and BigGAN as the data generator for ATOL, namely, *ATOL-StyleGAN* and *ATOL-BigGAN* (*ATOL-S* and *ATOL-B* for short), which are one of the most popular generative models.

Table 16: Performance comparisons with different data generators on CIFAR-10 and CIFAR-100 ↓ (or ↑) indicates smaller (or larger) values are preferred; a bold font indicates the best results.

| Method | SVHN | | LSUN-Crop | | LSUN-Resize | | iSUN | | Texture | | Places365 | | Average | |
|--------|------|------|-----------|------|-------------|------|------|------|---------|------|-----------|------|---------|------|
| | FPR95↓ | AUROC↑ | FPR95↓ | AUROC↑ | FPR95↓ | AUROC↑ | FPR95↓ | AUROC↑ | FPR95↓ | AUROC↑ | FPR95↓ | AUROC↑ | FPR95↓ | AUROC↑ |
| CIFAR-10 | | | | | | | | | | | | | | |
| ATOL | 20.60 | 96.03 | 1.45 | 99.59 | 5.20 | 98.78 | 5.00 | 98.76 | 26.05 | 95.03 | 27.55 | 94.33 | 14.31 | 97.09 |
| ATOL-S | 21.40 | 94.66 | 1.95 | 99.43 | 2.20 | 99.44 | 2.15 | 99.45 | 23.85 | 93.96 | 30.35 | 92.84 | 13.65 | 96.63 |
| ATOL-B | 12.75 | 96.92 | 4.60 | 98.92 | 0.65 | 99.78 | 0.55 | 99.83 | 10.25 | 97.12 | 22.85 | 94.80 | **8.61** | **97.90** |
| CIFAR-100 | | | | | | | | | | | | | | |
| ATOL | 70.85 | 84.70 | 13.45 | 97.52 | 51.85 | 90.12 | 55.80 | 89.02 | 63.10 | 83.37 | 75.30 | 78.86 | 55.06 | 87.26 |
| ATOL-S | 69.95 | 80.95 | 17.00 | 96.89 | 16.35 | 96.95 | 22.75 | 95.11 | 58.55 | 83.35 | 74.45 | 77.35 | 43.18 | 88.43 |
| ATOL-B | 54.65 | 89.69 | 43.95 | 91.27 | 7.80 | 98.57 | 9.60 | 98.13 | 37.45 | 89.51 | 66.90 | 80.82 | **36.72** | **91.33** |

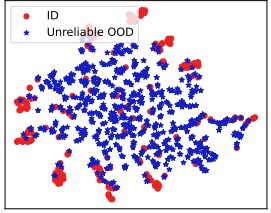
(a) Mistaken OOD Generation

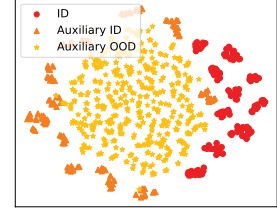
(b) ATOL without Alignment

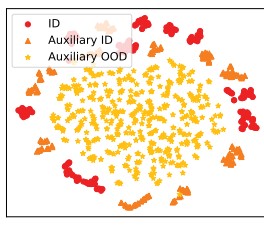
(c) ATOL

Figure 3: The t-SNE Visualization of empirical embedding feature distribution of ATOL training on CIFAR-10 dataset. The red circle represents the real ID data, the blue star represents the unreliable OOD generated data, the orange triangle represents auxiliary ID data, and the yellow star represents auxiliary OOD data. We qualitatively illustrate the results about mistaken OOD generation, ATOL without alignment and our ATOL.

## E.9 Visualization of Embedding Features

Qualitatively, to understand how our method helps the predictor to learn, we exploit t-SNE [61] to show the embedding distributions of real and auxiliary data. The embedding features are extracted from the penultimate layer of a WRN-40-2 model trained on CIFAR-10. In Figure 3(a), the mistaken OOD generation leads to an overlap between ID samples and the unreliable OOD samples in the embedding space. In contrast, our ATOL can make the predictor learn to discern the ID and OOD cases as in 3(b), relieving the mistaken OOD generation by a large margin. However, the auxiliary ID distribution is far from the real ID distribution, which indicates that the OOD detection capability in the auxiliary task cannot benefit the predictor in real OOD detection. Such a distribution shift results in poor OOD detection performance in the real task (cf. Section 5.3). As shown in Figure 3(c), we further align the real ID and auxiliary ID distribution in embedding space, where the features of auxiliary ID data are consistent with real ID data. Further, the auxiliary OOD data is observably separate from the real ID data, which proves that ATOL can benefit the predictor learning from the reliable OOD data, thereby providing strong flexibility and generality.

## E.10 Hardware Configurations

All experiments are realized by Pytorch 1.11 with CUDA 12.0, using machines equipped with NVIDIA Tesla A100 GPUs.

## F Further Experiments

### F.1 CIFAR Benchmarks

In this section, We compare our method with advanced OOD detection methods besides the data generation-based methods, including MSP [15], ODIN [35], Mahalanobis [32], Free Energy [36], MaxLogit [17], ReAct [56], ViM [64], KNN [57], Watermark [68], ASH [9], BoundaryGAN [31], ConfGAN [55], ManifoldGAN [62],CSI [58], G2D [50], CMG [67], LogitNorm [71], VOS [10],

Table 17: Comparison of ATOL and advanced methods on CIFAR-10 dataset. All methods are trained on ID data only, without using outlier data. ↓ (or ↑) indicates smaller (or larger) values are preferred; a bold font indicates the best results.

| Method | SVHN | | LSUN-Crop | | LSUN-Resize | | iSUN | | Texture | | Places365 | | Average | |
|---|---|---|---|---|---|---|---|---|---|---|---|---|---|---|
| | FPR95↓ | AUROC↑ | FPR95↓ | AUROC↑ | FPR95↓ | AUROC↑ | FPR95↓ | AUROC↑ | FPR95↓ | AUROC↑ | FPR95↓ | AUROC↑ | FPR95↓ | AUROC↑ |
| OOD Scoring Functions | | | | | | | | | | | | | | |
| MSP | 49.45 | 91.46 | 26.40 | 96.27 | 51.80 | 91.56 | 54.40 | 90.41 | 59.60 | 88.41 | 58.50 | 88.32 | 50.03 | 91.07 |
| ODIN | 29.16 | 91.52 | 12.84 | 96.29 | 32.53 | 90.19 | 42.27 | 89.79 | 44.51 | 88.74 | 34.54 | 90.25 | 32.64 | 91.13 |
| Mahalanobis | 13.33 | 97.57 | 39.56 | 94.10 | 43.21 | 93.14 | 43.55 | 92.80 | 15.46 | 97.18 | 68.23 | 84.69 | 37.23 | 93.25 |
| Free Energy | 37.00 | 90.29 | 5.85 | 98.93 | 30.35 | 93.88 | 33.00 | 92.61 | 52.15 | 85.72 | 42.15 | 89.25 | 33.42 | 91.78 |
| MaxLogit | 36.60 | 90.36 | 6.55 | 98.82 | 30.50 | 93.40 | 33.15 | 92.54 | 51.75 | 85.84 | 42.55 | 89.21 | 33.52 | 91.76 |
| ReAct | 50.73 | 86.36 | 6.48 | 98.70 | 29.23 | 94.59 | 36.53 | 92.92 | 59.08 | 85.16 | 42.76 | 90.24 | 37.47 | 91.33 |
| ViM | 56.97 | 89.77 | 49.96 | 91.43 | 63.54 | 88.15 | 62.20 | 88.47 | 45.20 | 91.76 | 47.86 | 91.45 | 54.29 | 90.17 |
| KNN | 31.29 | 95.01 | 26.84 | 95.33 | 25.89 | 95.36 | 29.48 | 94.28 | 41.21 | 92.08 | 44.02 | 90.47 | 33.12 | 93.76 |
| KNN+ | 8.98 | 98.33 | 7.94 | 97.90 | 18.67 | 96.92 | 23.55 | 94.65 | 16.57 | 96.72 | 36.33 | 93.98 | 18.67 | 96.42 |
| Watermark | 16.80 | 96.89 | 13.30 | 97.74 | 12.50 | 97.86 | 12.95 | 97.73 | 32.20 | 93.87 | 34.20 | 93.63 | 20.33 | 96.29 |
| ASH | 50.73 | 86.36 | 6.48 | 98.70 | 29.23 | 94.59 | 36.53 | 92.92 | 59.08 | 85.16 | 42.76 | 90.24 | 37.47 | 91.33 |
| OOD Training Strategies | | | | | | | | | | | | | | |
| BoundaryGAN | 86.15 | 79.48 | 19.05 | 96.50 | 41.75 | 92.18 | 46.35 | 90.63 | 70.15 | 78.71 | 70.15 | 81.25 | 55.60 | 86.46 |
| ConfGAN | 56.75 | 87.56 | 7.95 | 98.26 | 14.70 | 97.02 | 17.65 | 96.72 | 40.25 | 90.25 | 52.10 | 88.23 | 31.57 | 93.01 |
| ManifoldGAN | 26.20 | 94.51 | 5.05 | 98.84 | 27.25 | 95.22 | 30.70 | 93.94 | 32.05 | 91.06 | 38.85 | 90.95 | 26.68 | 94.09 |
| CSI | 20.48 | 96.63 | 1.88 | 99.55 | 6.18 | 98.78 | 5.49 | 98.99 | 21.07 | 96.27 | 33.73 | 93.68 | 14.80 | 97.31 |
| G2D | 6.10 | 98.63 | 8.30 | 98.41 | 45.35 | 89.48 | 45.25 | 88.96 | 36.80 | 88.63 | 49.20 | 86.34 | 31.83 | 91.74 |
| LogitNorm | 6.84 | 98.58 | 1.58 | 99.53 | 18.85 | 96.94 | 20.79 | 96.58 | 26.64 | 94.87 | 30.38 | 93.85 | 17.51 | 96.73 |
| CMG | 41.70 | 92.48 | 50.35 | 89.35 | 37.80 | 94.24 | 35.60 | 94.75 | 38.70 | 92.12 | 34.95 | 94.07 | 39.83 | 92.83 |
| VOS | 46.15 | 93.69 | 3.30 | 99.11 | 41.80 | 94.20 | 48.10 | 93.31 | 57.85 | 88.33 | 61.25 | 87.54 | 43.08 | 92.03 |
| NPOS | 36.55 | 93.30 | 9.98 | 98.03 | 21.87 | 95.60 | 28.93 | 94.25 | 52.83 | 85.74 | 39.56 | 89.71 | 31.62 | 92.77 |
| CIDER | 5.86 | 98.36 | 7.35 | 98.50 | 47.58 | 93.64 | 47.15 | 93.60 | 28.04 | 94.79 | 41.10 | 91.03 | 29.51 | 94.99 |
| ATOL | 20.60 | 96.03 | 1.45 | 99.59 | 5.20 | 98.78 | 5.00 | 98.76 | 26.05 | 95.03 | 27.55 | 94.33 | 14.31 | 97.09 |
| ATOL-S | 21.40 | 94.66 | 1.95 | 99.43 | 2.20 | 99.44 | 2.15 | 99.45 | 23.85 | 93.96 | 30.35 | 92.84 | 13.65 | 96.63 |
| ATOL-B | 12.75 | 96.92 | 4.60 | 98.92 | 0.65 | 99.78 | 0.55 | 99.83 | 10.25 | 97.12 | 22.85 | 94.80 | **8.61** | **97.90** |

NPOS [60] and CIDER [44]. For clarity, we divide the baseline methods into two categories: OOD scoring functions and OOD training strategies, referring to Appendix B.

We summarize the main experiments in Table 17-18 on CIFAR benchmarks for common OOD detection compared with the advanced OOD detection methods. For a fair comparison, all the methods only use ID data without using real OOD datasets. We show that ATOL can achieve superior OOD detection performance on average for the evaluation metrics of FPR95 and AUROC, outperforming the competitive rivals by a large margin. Specifically, We incorporate the auxiliary OOD detection task to benefit the predictor learn the OOD knowledge without accessing to the real OOD data, which significantly improve the OOD detection performance.

Compared with the best baseline KNN+ [57], ATOL reduces the FPR95 from 33.12% to 8.61% on CIFAR-10 and from 51.75% to 8.61% on CIFAR-100. Moreover, for the previous works that adopt similar concepts in synthesize the boundary samples as outliers in the embedding space, i.e., VOS [10] and NPOS [60], our ATOL also reveals better results, with 28.76% and 17.31% improvements on the CIFAR-10 dataset and 11.27% and 7.20% improvements on the CIFAR-100 dataset w.r.t. FPR95. It indicates that our data generation strategy can lead a better OOD learning compared with the synthesizing features strategies. Note that, we can further benefit our ATOL from the latest progress in OOD scoring, improving the performance of our method in OOD detection (cf. Appendix F.6).

## F.2  ImageNet Benchmarks

Table 19 lists the detailed experiments on the ImageNet benchmark. The baselines are the same as what we described in Section F.1. Our ATOL achieves superior performance on average against all the considered baselines. Further, for the cases with iNaturalist and Places365, which are believed to be the challenging OOD datasets on the ImageNet situation, our ATOL also achieves considerable improvements against all other advanced methods. We highlight that ATOL outperforms the best baseline (i.e., CSI) by 5.55% in FPR95, and ATOL is also simpler to use and implement than CSI, which relies on sophisticated data augmentations and ensemble in testing. Overall, it demonstrates that our ATOL can also work well for challenging detection scenarios with extremely large semantic space and complex data patterns.

## F.3  Hard OOD Detection

Besides the above test OOD datasets, we also consider hard OOD scenarios [58], of which the test OOD data are very similar to that of the ID cases in style. Following the common setup [57] with the CIFAR-10 dataset being the ID case, we evaluate our ATOL on three hard OOD datasets, namely, LSUN-Fix [76], ImageNet-Fix [8], and CIFAR-100. We compare our ATOL with several works reported performing well in hard OOD detection, including KNN [57], ASH [9] and CSI [58], where

Table 18: Comparison of ATOL and advanced methods on CIFAR-100 dataset. All methods are trained on ID data only, without using outlier data. ↓ (or ↑) indicates smaller (or larger) values are preferred; a bold font indicates the best results.

| Method | SVHN | | LSUN-Crop | | LSUN-Resize | | iSUN | | Texture | | Places365 | | Average | |
|---|---|---|---|---|---|---|---|---|---|---|---|---|---|---|
| | FPR95↓ | AUROC↑ | FPR95↓ | AUROC↑ | FPR95↓ | AUROC↑ | FPR95↓ | AUROC↑ | FPR95↓ | AUROC↑ | FPR95↓ | AUROC↑ | FPR95↓ | AUROC↑ |
| OOD Scoring Functions | | | | | | | | | | | | | | |
| MSP | 83.95 | 72.80 | 59.30 | 85.33 | 80.45 | 76.51 | 81.85 | 76.19 | 83.00 | 74.01 | 82.10 | 74.39 | 78.44 | 76.54 |
| ODIN | 70.75 | 72.57 | 46.20 | 85.31 | 63.38 | 76.55 | 60.23 | 74.83 | 60.31 | 76.96 | 61.61 | 77.72 | 60.41 | 77.32 |
| Mahalanobis | 48.66 | 87.40 | 98.30 | 57.28 | 38.27 | 90.77 | 41.91 | 89.38 | 45.37 | 87.65 | 95.11 | 65.45 | 61.26 | 79.65 |
| Free Energy | 85.55 | 74.53 | 23.90 | 95.53 | 77.60 | 80.22 | 80.30 | 79.07 | 79.95 | 76.24 | 79.15 | 76.37 | 71.07 | 80.33 |
| MaxLogit | 84.40 | 74.72 | 28.20 | 94.77 | 77.25 | 80.19 | 79.45 | 79.14 | 79.05 | 76.34 | 79.25 | 76.47 | 71.27 | 80.27 |
| ReAct | 51.22 | 77.77 | 21.91 | 95.67 | 68.54 | 77.95 | 66.82 | 78.06 | 58.81 | 79.54 | 69.22 | 76.27 | 56.09 | 80.88 |
| ViM | 72.32 | 82.92 | 74.03 | 81.57 | 84.89 | 77.03 | 84.15 | 76.69 | 33.51 | 91.72 | 64.17 | 79.57 | 68.78 | 81.58 |
| KNN | 42.39 | 92.26 | 59.46 | 79.88 | 59.46 | 88.85 | 59.89 | 87.48 | 48.30 | 88.90 | 81.27 | 74.82 | 59.99 | 85.37 |
| KNN+ | 49.73 | 88.06 | 59.27 | 82.20 | 31.94 | 93.81 | 37.11 | 91.86 | 48.30 | 87.96 | 84.16 | 71.96 | 51.75 | 85.98 |
| Watermark | 84.95 | 75.04 | 73.15 | 85.74 | 72.95 | 85.79 | 71.95 | 85.47 | 71.95 | 81.82 | 79.25 | 77.48 | 75.70 | 81.89 |
| ASH | 65.63 | 87.44 | 18.90 | 96.77 | 76.81 | 80.53 | 79.26 | 79.59 | 72.71 | 80.54 | 81.72 | 74.84 | 65.84 | 83.29 |
| OOD Training Strategies | | | | | | | | | | | | | | |
| BoundaryGAN | 84.50 | 71.00 | 53.55 | 88.38 | 74.80 | 78.87 | 77.90 | 77.60 | 86.00 | 66.80 | 83.55 | 72.07 | 76.72 | 75.79 |
| ConfGAN | 88.30 | 72.04 | 39.35 | 92.01 | 77.85 | 80.26 | 79.70 | 79.47 | 79.65 | 71.27 | 84.30 | 70.99 | 74.86 | 77.67 |
| ManifoldGAN | 81.65 | 74.51 | 23.90 | 91.52 | 80.20 | 73.16 | 81.80 | 74.07 | 76.35 | 76.31 | 81.30 | 74.84 | 73.54 | 77.40 |
| CSI | 62.96 | 84.75 | 61.84 | 85.82 | 96.47 | 49.28 | 95.91 | 52.98 | 78.30 | 71.25 | 85.00 | 71.45 | 80.08 | 85.23 |
| G2D | 45.40 | 88.56 | 42.40 | 91.38 | 85.80 | 73.22 | 84.70 | 74.66 | 83.95 | 72.18 | 82.10 | 74.19 | 70.73 | 79.03 |
| LogitNorm | 41.37 | 92.78 | 14.57 | 97.20 | 87.96 | 67.88 | 89.15 | 65.74 | 69.66 | 78.87 | 78.00 | 78.59 | 63.45 | 80.18 |
| CMG | 73.05 | 82.12 | 86.50 | 72.07 | 81.45 | 77.80 | 82.70 | 76.85 | 80.30 | 73.86 | 73.65 | 82.39 | 79.60 | 77.51 |
| VOS | 78.06 | 82.59 | 40.40 | 92.90 | 83.47 | 70.82 | 85.77 | 70.20 | 82.46 | 77.22 | 82.31 | 75.47 | 75.41 | 78.20 |
| NPOS | 66.09 | 87.59 | 31.70 | 95.11 | 63.34 | 80.98 | 62.59 | 80.27 | 74.76 | 80.45 | 77.86 | 80.65 | 62.72 | 84.17 |
| CIDER | 52.21 | 88.44 | 46.88 | 90.18 | 52.23 | 89.89 | 47.57 | 89.91 | 84.67 | 70.62 | 84.67 | 71.82 | 61.37 | 83.48 |
| ATOL | 75.50 | 81.50 | 10.05 | 98.15 | 55.35 | 87.46 | 56.75 | 87.64 | 64.90 | 83.32 | 70.55 | 79.86 | 55.52 | 86.32 |
| ATOL-S | 69.95 | 80.95 | 17.00 | 96.89 | 16.35 | 96.95 | 22.75 | 95.11 | 58.55 | 83.35 | 74.45 | 77.35 | 43.18 | 88.43 |
| ATOL-B | 54.65 | 89.69 | 43.95 | 91.27 | 7.80 | 98.57 | 9.60 | 98.13 | 37.45 | 89.51 | 66.90 | 80.82 | **36.72** | **91.33** |

Table 19: Comparison of ATOL and advanced methods on ImageNet dataset. All methods are trained on ID data only, without using outlier data. ↓ (or ↑) indicates smaller (or larger) values are preferred; a bold font indicates the best results.

| Method | iNaturalist | | SUN | | Places365 | | Texture | | Average | |
|---|---|---|---|---|---|---|---|---|---|---|
| | FPR95↓ | AUROC↑ | FPR95↓ | AUROC↑ | FPR95↓ | AUROC↑ | FPR95↓ | AUROC↑ | FPR95↓ | AUROC↑ |
| OOD Scoring Functions | | | | | | | | | | |
| MSP | 64.25 | 78.49 | 87.47 | 68.61 | 85.56 | 72.29 | 63.60 | 79.28 | 75.22 | 74.67 |
| ODIN | 63.85 | 77.78 | 89.98 | 61.80 | 88.00 | 67.17 | 67.87 | 77.40 | 77.43 | 71.04 |
| Mahalanobis | 95.90 | 60.56 | 95.42 | 45.33 | 98.90 | 44.65 | 55.80 | 84.60 | 86.50 | 58.78 |
| Free Energy | 61.14 | 77.65 | 90.20 | 61.59 | 89.88 | 63.78 | 57.61 | 77.61 | 74.71 | 70.16 |
| MaxLogit | 57.41 | 81.85 | 83.91 | 70.63 | 81.36 | 73.54 | 63.42 | 76.18 | 71.53 | 75.55 |
| ReAct | 55.47 | 81.20 | 66.81 | 82.59 | 63.71 | 85.11 | 46.33 | 90.30 | 58.08 | 84.80 |
| ViM | 85.44 | 77.16 | 89.50 | 75.00 | 87.73 | 78.25 | 38.47 | 90.55 | 75.29 | 80.24 |
| KNN | 65.40 | 83.73 | 75.62 | 77.33 | 79.20 | 74.34 | 40.80 | 86.45 | 64.75 | 80.91 |
| KNN+ | 61.48 | 80.35 | 74.04 | 77.33 | 63.53 | 82.95 | 33.48 | 90.53 | 58.13 | 82.79 |
| Watermark | 67.36 | 80.60 | 72.64 | 79.55 | 79.20 | 70.80 | 81.74 | 83.46 | 75.24 | 78.61 |
| ASH | 80.00 | 67.39 | 92.20 | 58.97 | 87.01 | 68.10 | 70.18 | 69.61 | 82.35 | 66.02 |
| OOD Training Strategies | | | | | | | | | | |
| BoundaryGAN | 83.36 | 68.68 | 90.75 | 63.84 | 87.66 | 67.50 | 80.16 | 68.12 | 85.48 | 66.78 |
| ConfGAN | 72.67 | 78.29 | 80.73 | 73.88 | 77.40 | 77.24 | 68.74 | 78.74 | 74.88 | 77.03 |
| ManifoldGAN | 72.50 | 78.08 | 84.40 | 72.67 | 82.85 | 74.90 | 50.28 | 85.28 | 72.50 | 78.08 |
| CSI | 64.24 | 85.47 | 53.92 | 95.30 | 58.68 | 93.00 | 52.57 | 91.13 | 57.35 | **91.22** |
| G2D | 73.44 | 78.11 | 81.28 | 74.05 | 77.18 | 77.33 | 67.82 | 79.17 | 74.93 | 77.16 |
| LogitNorm | 76.96 | 77.94 | 75.91 | 79.07 | 72.65 | 81.44 | 71.63 | 80.15 | 74.29 | 79.65 |
| CMG | 71.33 | 78.78 | 80.88 | 73.93 | 75.77 | 77.58 | 63.83 | 80.57 | 72.95 | 77.63 |
| VOS | 87.52 | 72.45 | 83.85 | 74.76 | 79.97 | 77.26 | 72.91 | 82.71 | 81.06 | 76.79 |
| NPOS | 74.74 | 77.43 | 83.09 | 73.73 | 78.23 | 76.91 | 56.10 | 84.37 | 73.04 | 78.11 |
| CIDER | 79.22 | 67.27 | 84.82 | 74.34 | 77.39 | 78.77 | 19.80 | 95.16 | 65.31 | 78.89 |
| ATOL | 60.98 | 79.53 | 73.90 | 79.97 | 58.48 | 86.97 | 13.85 | 96.80 | **51.80** | 85.82 |

the results are summarized in Table 20. As we can see, our ATOL can beat these advanced methods across all the considered datasets, even for the challenging CIFAR-10 vs. CIFAR-100 setting.

## F.4 Comparison of ATOL and Outlier Exposure

This section compares ATOL and OE on CIFAR and ImageNet benchmarks in Tables 21-22. As we can see, our ATOL shows comparable performance with outlier exposure, meanwhile eliminating the reliance on real OOD data. Such outstanding performances are sufficient to verify that ATOL can make the predictor learn from the auxiliary OOD detection task as effectively as learning from the real OOD data. Note that, on the ImageNet benchmark, OE reveals inferior performance compared to ATOL since the real OOD data collected from the realistic scenarios inevitably overlap with the large-scale ID data to some extent, while getting pure and efficient OOD data is labor-intensive and inflexible. The results demonstrate the drawbacks of the methods using real OOD data. In contrast, our ATOL does not suffer from this issue and can benefit from the auxiliary OOD detection task for

Table 20: Comparison of ATOL and advanced methods in hard OOD detection. ↓ (or ↑) indicates smaller (or larger) values are preferred; a bold font indicates the best results in a column.

| Methods | LSUN-Fix | | ImageNet-Fix | | CIFAR-100 | |
|---|---|---|---|---|---|---|
| | FPR95 ↓ | AUROC ↑ | FPR95 ↓ | AUROC ↑ | FPR95 ↓ | AUROC ↑ |
| CSI | 39.79 | 93.63 | 37.47 | 93.93 | 45.64 | 87.64 |
| KNN | 42.01 | 91.98 | 40.41 | 92.33 | 49.22 | 89.91 |
| ASH | 45.12 | 89.72 | 42.56 | 89.99 | 50.45 | 87.09 |
| ATOL | **22.25** | **95.89** | **24.20** | **94.99** | **42.80** | **91.19** |

real OOD detection. Moreover, the better results of our ATOL over OE verify the effectiveness of OOD learning only based on ID data, which can draw more attention from the community toward OOD learning with ID data.

Table 21: Comparison of ATOL and Outlier Exposure on CIFAR-10 and CIFAR-100. ↓ (or ↑) indicates smaller (or larger) values are preferred; a bold font indicates the best results in a column.

| Scores | SVHN | | LSUN-Crop | | LSUN-Resize | | iSUN | | Texture | | Places365 | | Average | |
|---|---|---|---|---|---|---|---|---|---|---|---|---|---|---|
| | FPR95↓ | AUROC↑ | FPR95↓ | AUROC↑ | FPR95↓ | AUROC↑ | FPR95↓ | AUROC↑ | FPR95↓ | AUROC↑ | FPR95↓ | AUROC↑ | FPR95↓ | AUROC↑ |
| | | | | | | | | CIFAR-10 | | | | | | |
| ATOL | 20.60 | 96.03 | 1.45 | 99.59 | 5.20 | 98.78 | 5.00 | 98.76 | 26.05 | 95.03 | 27.55 | 94.33 | 14.31 | 97.09 |
| ATOL-S | 21.40 | 94.66 | 1.95 | 99.43 | 2.20 | 99.44 | 2.15 | 99.45 | 23.85 | 93.96 | 30.35 | 92.84 | 13.65 | 96.63 |
| ATOL-B | 12.75 | 96.92 | 4.60 | 98.92 | 0.65 | 99.78 | 0.55 | 99.83 | 10.25 | 97.12 | 22.85 | 94.80 | **8.61** | **97.90** |
| OE | 20.35 | 96.75 | 2.20 | 99.52 | 0.95 | 99.72 | 0.85 | 99.76 | 17.15 | 96.79 | 23.05 | 95.64 | 10.76 | 97.03 |
| | | | | | | | | CIFAR-100 | | | | | | |
| ATOL | 75.50 | 81.50 | 10.05 | 98.15 | 55.35 | 87.46 | 56.75 | 87.64 | 64.90 | 83.32 | 70.55 | 79.86 | 55.52 | 86.32 |
| ATOL-S | 69.95 | 80.95 | 17.00 | 96.89 | 16.35 | 96.95 | 22.75 | 95.11 | 58.55 | 83.35 | 74.45 | 77.35 | 43.18 | 88.43 |
| ATOL-B | 54.65 | 89.69 | 43.95 | 91.27 | 7.80 | 98.57 | 9.60 | 98.13 | 37.45 | 89.51 | 66.90 | 80.82 | **36.72** | **91.33** |
| OE | 75.10 | 80.69 | 24.95 | 95.11 | 20.05 | 95.36 | 25.45 | 93.94 | 48.15 | 87.94 | 43.25 | 88.55 | 39.49 | 90.26 |

Table 22: Comparison of ATOL and Outlier Exposure on ImageNet. ↓ (or ↑) indicates smaller (or larger) values are preferred; a bold font indicates the best results in a column.

| Scores | iNaturalist | | SUN | | Places365 | | Texture | | Average | |
|---|---|---|---|---|---|---|---|---|---|---|
| | FPR95↓ | AUROC↑ | FPR95↓ | AUROC↑ | FPR95↓ | AUROC↑ | FPR95↓ | AUROC↑ | FPR95↓ | AUROC↑ |
| ATOL | 60.98 | 79.53 | 73.90 | 79.97 | 58.48 | 86.97 | 13.85 | 96.80 | **51.80** | **85.82** |
| OE | 78.31 | 75.23 | 80.10 | 76.55 | 70.41 | 81.78 | 66.38 | 82.04 | 73.80 | 78.90 |

## F.5 Model Trained from Scratch

In this section, we provide the implementation details and the experimental results for ATOL trained from scratch. We train the WRN-40-2 model on CIFAR-10. For training, ATOL is run for 100 epochs with an initial learning rate of 0.1 and cosine decay. The batch size is 64 for ID cases and 256 for OOD cases. Going beyond fine-tuning with the pre-trained model, we show that ATOL is also applicable and effective when training from scratch. Here, we further introduce the *area under the precision-recall curve* (AUPR) to evaluate the OOD detection performance. The table 23 showcases the performance of ATOL trained on the CIFAR-10 dataset, where the promising performance of ATOL still holds.

## F.6 ATOL with different scoring functions

To further verify the generality and the effectiveness of ATOL, we test ATOL with three representative OOD scoring functions, namely, MSP [15], Free energy [36], and MaxLogit [17]. Regarding all the cases with different scoring functions, our ATOL always achieves good performance, demonstrating that our proposal can genuinely make the predictor learn from OOD knowledge for OOD detection. Further, comparing the results across different scoring strategies, we observe that using the MaxLogit scoring leads to better results than the MSP scoring. Therefore, we choose the MaxLogit in ATOL.

## F.7 Performance and Efficiency Comparison on Advanced Approaches

In the main context, our primary focus is on comparing data generation-based approaches. Moreover, we also compare the training time for a set of representative methods on CIFAR-100 (similar results on

Table 23: Performance of ATOL from scratch on CIFAR-10

| ATOL from scratch | FPR95 ↓ | AUROC ↑ | AUPR ↑ |
|---|---|---|---|
| SVHN | 18.85 | 96.63 | 99.30 |
| LSUN-Crop | 2.20 | 99.51 | 99.90 |
| LSUN-Resize | 6.70 | 97.88 | 99.56 |
| iSUN | 7.25 | 97.83 | 99.55 |
| Texture | 26.40 | 94.77 | 98.62 |
| Places365 | 30.35 | 92.53 | 97.95 |
| Average | 15.29 | 96.52 | 99.15 |

Table 24: Performance with different OOD scoring functions on CIFAR-10 and CIFAR-100

| Scores | SVHN | | LSUN-Crop | | LSUN-Resize | | iSUN | | Texture | | Places365 | | Average | |
|---|---|---|---|---|---|---|---|---|---|---|---|---|---|---|
| | FPR95 ↓ | AUROC ↑ | FPR95 ↓ | AUROC ↑ | FPR95 ↓ | AUROC ↑ | FPR95 ↓ | AUROC ↑ | FPR95 ↓ | AUROC ↑ | FPR95 ↓ | AUROC ↑ | FPR95 ↓ | AUROC ↑ |
| CIFAR-10 | | | | | | | | | | | | | | |
| MSP | 29.10 | 94.57 | 1.45 | 99.58 | 6.95 | 98.60 | 6.75 | 98.63 | 22.45 | 94.78 | 27.30 | 93.27 | 15.67 | 96.57 |
| Energy | 16.45 | 96.27 | 1.25 | 99.60 | 3.35 | 99.29 | 3.70 | 99.22 | 26.00 | 93.75 | 35.70 | 92.14 | 14.41 | 96.71 |
| MaxLogit | 21.15 | 95.97 | 1.20 | 99.61 | 4.40 | 98.87 | 5.40 | 98.75 | 24.75 | 95.18 | 25.40 | 94.74 | 13.72 | 97.19 |
| CIFAR-100 | | | | | | | | | | | | | | |
| MSP | 69.70 | 83.88 | 16.75 | 96.77 | 56.60 | 88.55 | 58.95 | 87.55 | 64.60 | 82.14 | 77.00 | 78.09 | 57.27 | 86.16 |
| Energy | 36.00 | 94.05 | 8.85 | 98.12 | 38.15 | 89.07 | 42.25 | 87.77 | 66.65 | 81.37 | 75.40 | 77.01 | 44.55 | 87.90 |
| MaxLogit | 70.85 | 84.70 | 13.45 | 97.52 | 51.85 | 90.12 | 55.80 | 89.02 | 63.10 | 83.37 | 75.30 | 78.86 | 55.06 | 87.26 |

CIFAR-10), summarizing the results in the following table. As we can see, our method demonstrates promising performance improvement over other methods with acceptable computational resources.

## F.8 Mean and Standard Deviation

This section demonstrates the effectiveness of our ATOL by validating the experiments using five individual trails (random seeds) on the CIFAR benchmarks. Along with the individual findings, we also summarize the mean performance and standard deviation for all of the trials for CIFAR-10, CIFAR-100 and ImageNet. The experimental results are summarized in Tables 27-28. As we can see, our ATOL can result in better OOD detection performance and more stable performance over various ID dataset options.

## F.9 Visualization of generated images

In this section, we visualize some synthesized examples for intuitive demonstration. As demonstrated in the main paper, ATOL performs surprisingly well on CIFAR-10, CIFAR-100, and ImageNet. ATOL also enables us to generate visual results for intuitive inspection. For auxiliary ID cases, we sample noise from different Gaussian distributions from MoG with different means but the same standard deviation in latent space, representing different classes of the auxiliary ID data. For auxiliary OOD cases, we sample noise from a uniform distribution in latent space except for the region with high MoG density. The generated auxiliary data are visualized in Figure 4-9.

Except for the mistaken OOD generation on CIFAR datasets, we further visualize the mistaken OOD generation data of the advanced data generation-based methods on ImageNet dataset. Since the generators are trained on ID data and these selection procedures can make mistakes, one may wrongly select data with ID semantics as OOD cases (cf., Figure 10). As we can see, the increased difficulty in searching for the proper OOD-like data in ImageNet leads to more critical mistaken OOD generation.

Table 25: Performance with different OOD scoring functions on ImageNet.

| Scores | iNaturalist | | SUN | | Places365 | | Texture | | Average | |
|---|---|---|---|---|---|---|---|---|---|---|
| | FPR95 ↓ | AUROC ↑ | FPR95 ↓ | AUROC ↑ | FPR95 ↓ | AUROC ↑ | FPR95 ↓ | AUROC ↑ | FPR95 ↓ | AUROC ↑ |
| MSP | 61.16 | 80.38 | 73.37 | 77.65 | 61.73 | 83.63 | 22.91 | 93.55 | 54.79 | 83.80 |
| Energy | 62.15 | 80.49 | 72.13 | 82.15 | 62.69 | 85.78 | 31.10 | 92.73 | 57.02 | 85.29 |
| MaxLogit | 60.98 | 79.53 | 73.90 | 79.97 | 58.48 | 86.97 | 13.85 | 96.80 | 51.80 | 85.82 |

Table 26: Performance and efficiency comparison with advanced approaches on CIFAR-100. We report the per epoch training time (measured by seconds)

| Methods | FPR95 ↓ | AUROC ↑ | Training Time (s)↓ |
|---|---|---|---|
| CSI | 80.08 | 85.23 | 98.88 |
| LogitNorm | 63.45 | 80.18 | **25.16** |
| VOS | 75.41 | 78.20 | 38.97 |
| NPOS | 62.72 | 84.17 | 61.54 |
| ATOL | 55.22 | 87.24 | 58.33 |
| ATOL-S | 43.18 | 88.43 | 70.28 |
| ATOL-B | **36.72** | **91.33** | 65.33 |

Table 27: Performance of ATOL on CIFAR with 5 individual trails. ↓ (or ↑) indicates smaller (or larger) values are preferred; and a bold font indicates the best results in the corresponding column.

| Scores | SVHN | | LSUN-Crop | | LSUN-Resize | | iSUN | | Texture | | Places365 | | Average | |
|---|---|---|---|---|---|---|---|---|---|---|---|---|---|---|
| | FPR95 ↓ | AUROC ↑ | FPR95 ↓ | AUROC ↑ | FPR95 ↓ | AUROC ↑ | FPR95 ↓ | AUROC ↑ | FPR95 ↓ | AUROC ↑ | FPR95 ↓ | AUROC ↑ | FPR95 ↓ | AUROC ↑ |
| | | | | | | | CIFAR-10 | | | | | | | |
| #1 | 21.55 | 96.10 | 0.95 | 99.62 | 5.00 | 98.79 | 4.55 | 98.90 | 26.30 | 95.04 | 28.90 | 93.60 | 14.54 | 97.01 |
| #2 | 21.65 | 95.74 | 1.15 | 99.60 | 5.40 | 98.79 | 3.80 | 98.93 | 26.25 | 94.91 | 27.75 | 94.19 | 14.33 | 97.03 |
| #3 | 18.05 | 96.27 | 1.45 | 99.58 | 4.35 | 98.88 | 4.45 | 98.86 | 20.90 | 95.70 | 25.85 | 94.39 | 12.51 | 97.28 |
| #4 | 20.85 | 96.11 | 1.60 | 99.59 | 5.05 | 98.76 | 5.75 | 98.73 | 26.50 | 94.80 | 28.20 | 94.32 | 14.66 | 97.05 |
| #5 | 21.40 | 94.66 | 1.95 | 99.43 | 2.20 | 99.44 | 2.15 | 99.45 | 23.85 | 93.96 | 30.35 | 92.84 | 13.65 | 96.63 |
| mean | **20.70** | **95.77** | **1.42** | **99.56** | **4.4** | **98.93** | **4.14** | **98.97** | **24.75** | **94.88** | **28.21** | **93.86** | **13.93** | **97.00** |
| ± std | ± 1.35 | ± 0.58 | ± 0.34 | ± 0.06 | ± 1.15 | ± 0.25 | ± 1.17 | ± 0.24 | ± 2.16 | ± 0.55 | ± 1.47 | ± 0.58 | ± 0.79 | ± 0.20 |
| | | | | | | | CIFAR-100 | | | | | | | |
| #1 | 68.45 | 85.62 | 14.10 | 97.41 | 54.70 | 88.63 | 59.55 | 87.45 | 61.60 | 83.56 | 74.70 | 78.76 | 55.52 | 86.90 |
| #2 | 70.05 | 86.34 | 14.90 | 97.27 | 51.60 | 89.37 | 55.55 | 88.34 | 63.30 | 83.37 | 75.95 | 78.76 | 55.22 | 87.24 |
| #3 | 70.85 | 84.70 | 13.45 | 97.52 | 51.85 | 90.12 | 55.80 | 89.02 | 63.10 | 83.37 | 75.30 | 78.86 | 55.06 | 87.26 |
| #4 | 65.55 | 86.13 | 14.55 | 97.37 | 53.00 | 89.31 | 55.65 | 88.81 | 67.50 | 82.39 | 75.35 | 77.76 | 55.27 | 86.96 |
| #5 | 69.45 | 84.80 | 14.35 | 97.67 | 55.45 | 89.08 | 58.10 | 88.30 | 61.70 | 84.64 | 74.70 | 78.16 | 55.63 | 87.11 |
| mean | **68.86** | **85.52** | **14.27** | **97.44** | **53.32** | **89.30** | **56.92** | **88.38** | **63.44** | **83.46** | **75.19** | **78.46** | **55.34** | **87.09** |
| ± std | ± 1.83 | ± 0.67 | ± 0.48 | ± 0.13 | ± 1.52 | ± 0.48 | ± 1.61 | ± 0.54 | ± 2.14 | ± 0.71 | ± 0.46 | ± 0.42 | ± 0.20 | ± 0.14 |

# G   Broader Impacts and Limitations

**Broader impacts.**    This paper pioneers work on the problem of mistaken OOD generation in OOD detection, which is significant for the safety-critical applications of models with the rapid development of machine learning. Our method is proposed for the problem, relieving the this problem by a large margin and achieving superior performance. Due to data privacy and security, access of data is often challenging. In our method, we propose an auxiliary OOD detection task built upon the generator with ID knowledge to make the predictor learn to discern ID and OOD cases. Note that, the idea of the auxiliary task can be extended to other domains beyond OOD detection, which may result in the particular applications of other techniques.

**Limitations.**    First, our current realization of ATOL is relatively intricate, requiring training constraints on both the generator and the predictor. Further studies will explore more advanced conditions that can ease our realization and further reduce computing costs. Second, we observe that the diversity of generated data is closely related to the final performance (cf., Appendix E.8). However, in our current version, we do not consider diversity for the generator in either theories or algorithms, which will motivate our following exploration.

Table 28: Performance of ATOL on ImageNet with 5 individual trails. ↓ (or ↑) indicates smaller (or larger) values are preferred; and a bold font indicates the best results in the corresponding column.

| Scores | iNaturalist | | SUN | | Places365 | | Texture | | Average | |
|---|---|---|---|---|---|---|---|---|---|---|
| | FPR95 ↓ | AUROC ↑ | FPR95 ↓ | AUROC ↑ | FPR95 ↓ | AUROC ↑ | FPR95 ↓ | AUROC ↑ | FPR95 ↓ | AUROC ↑ |
| #1 | 60.98 | 79.53 | 73.90 | 79.97 | 58.48 | 86.97 | 13.85 | 96.80 | 51.80 | 85.82 |
| #2 | 62.87 | 78.82 | 73.95 | 80.48 | 58.67 | 87.14 | 13.95 | 96.79 | 52.36 | 85.80 |
| #3 | 61.50 | 78.99 | 73.53 | 80.28 | 58.73 | 87.09 | 13.95 | 96.79 | 51.93 | 85.79 |
| #4 | 58.07 | 80.10 | 72.32 | 81.09 | 59.41 | 86.70 | 15.20 | 96.56 | 51.25 | 86.12 |
| #5 | 61.70 | 78.77 | 73.06 | 80.11 | 58.80 | 87.20 | 13.95 | 96.79 | 51.88 | 85.72 |
| mean | **61.02** | **79.24** | **73.35** | **80.38** | **58.81** | **87.02** | **14.18** | **96.74** | **51.84** | **85.85** |
| ± std | ± 1.60 | ± 0.50 | ± 0.60 | ± 0.39 | ± 0.31 | ± 0.17 | ± 0.51 | ± 0.09 | ± 0.35 | ± 0.13 |

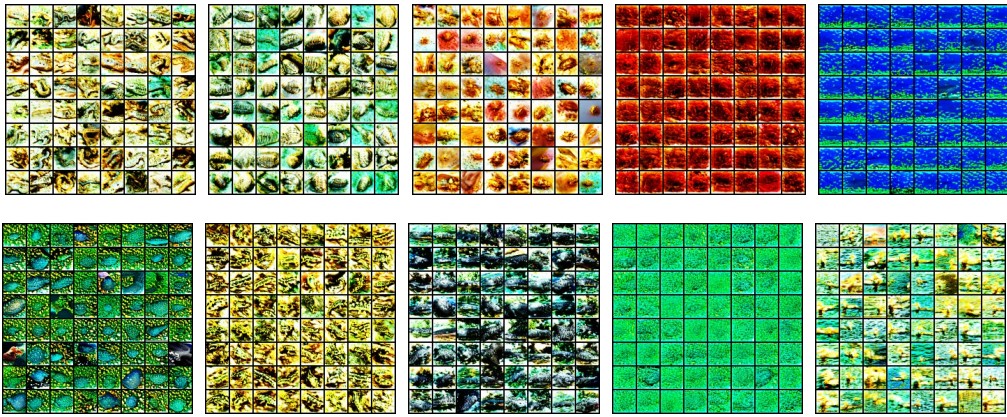

Figure 4: Generated auxiliary ID data on CIFAR-10 dataset. Each figure contains 64 images of each class.

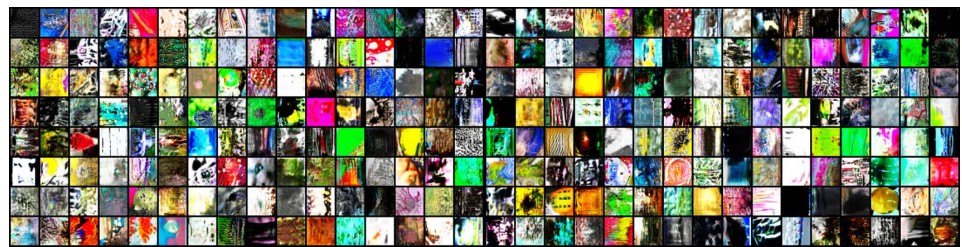

Figure 5: Generated auxiliary OOD data on CIFAR-10 dataset.

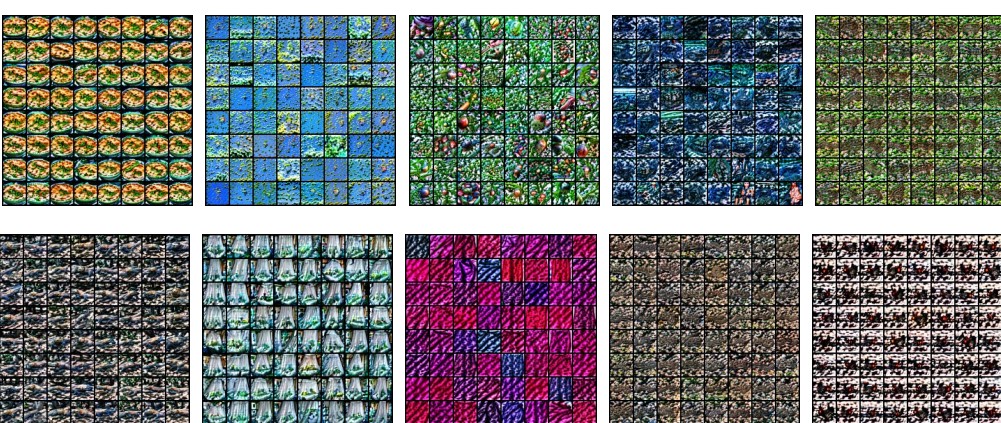

Figure 6: Generated auxiliary ID data on CIFAR-100 dataset. Each figure contains 64 images of each class. Due to space limitations, we only show 10 out of 100 classes.

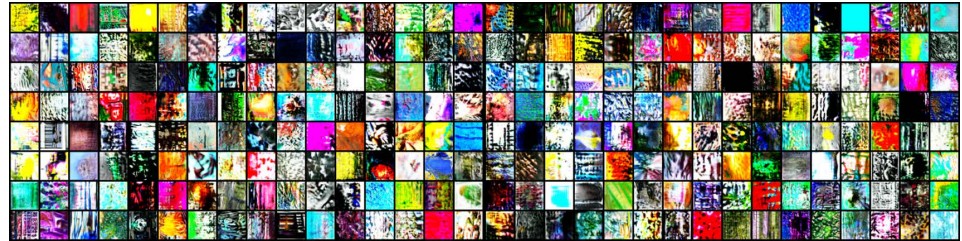

Figure 7: Generated auxiliary OOD data on CIFAR-100 dataset.

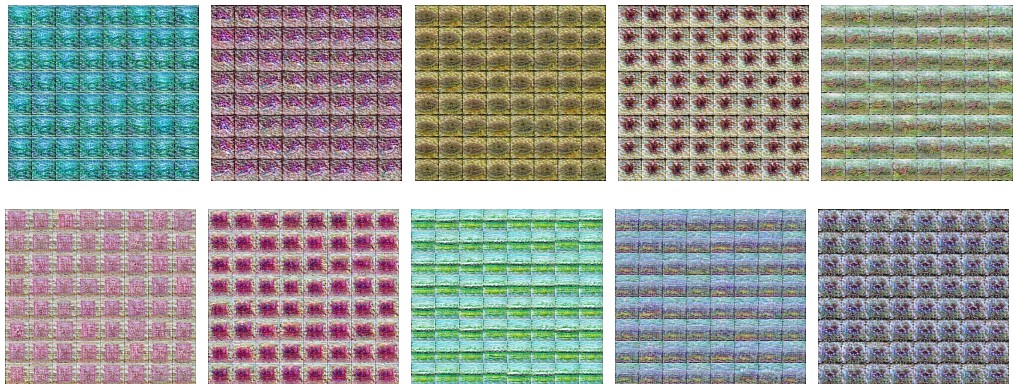

Figure 8: Generated auxiliary ID data on ImageNet dataset. Each figure contains 64 images of each class. Due to space limitations, we only show 10 out of 1000 classes.

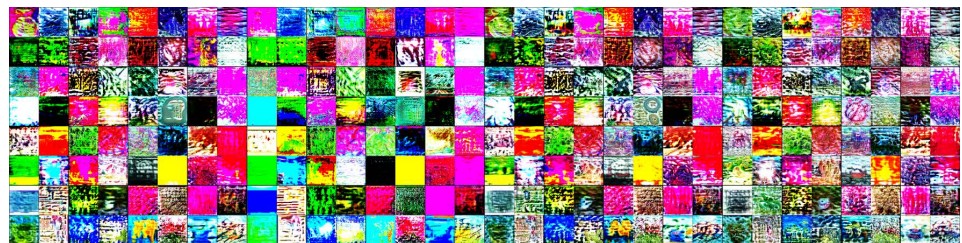

Figure 9: Generated auxiliary OOD data on ImageNet dataset.

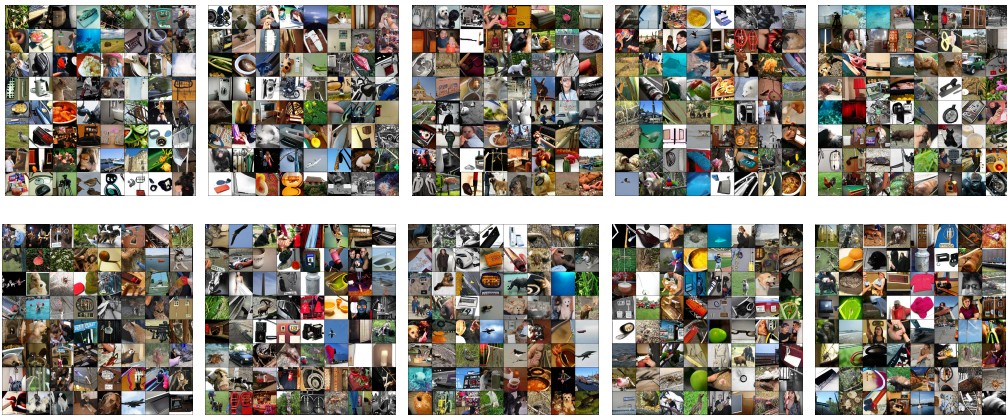

Figure 10: Mistaken OOD generation on ImageNet dataset. Each figure contains 64 images.