# OpenReview forum: "Out-of-distribution Detection Learning with Unreliable Out-of-distribution Sources"
_NeurIPS.cc/2023/Conference — NeurIPS 2023 poster_

### Official Review · Reviewer_iqYB · 2023-06-26

**Soundness:** 3 good
**Presentation:** 3 good
**Contribution:** 3 good
**Rating:** 5
**Confidence:** 3

**Summary:**

This paper contributes a new data-generation method to train a OOD detector. The paper proposes to first create an auxiliary task for a generator to generate in-distribution samples and OOD samples by using regions of disjoint latent space (equ 6-7). This can lead to disjoint support set by enforcing a distance preserving loss (equ 8). In order to transfer auxiliary task to real OOD detection, it suggests to use a contrastive learning to bring together generated in-dist samples and real in-dist samples. With these two settings, the OOD detector can learn from generated OOD samples.

**Strengths:**

This discussion is complete and clear. The idea of crafting the auxiliary task using generative model(s) is promising in general. By dividing the samples into high and low density regions, this method bypasses the issue of mistaken OOD samples.

**Weaknesses:**

The major concern is the performance of the generator (or possibly I miss that piece of information). While section 3 is convincing, in practice (section 4) how to guarantee high MoG density region has high concentration of in-distribution samples. This may happen when a generator does not perform well.

**Questions:**

$\bullet$ Can the author change the term predictor to OOD detector?

$\bullet$ While it is a good idea to generate in-dist and OOD samples using high MoG density region, does it also make the task too easy? I imagine that high MoG region samples may look like in-dist samples, while the rest may look like noise. Does the predictor can perform better if it can learn from more difficult (harder to discern) generated in-dist and generated OOD samples?

$\bullet$ Does this method consider the case when high MoG density region still have some OOD samples?

$\bullet$ The performance gain of the (OOD) predictor may be comparable to the generative model used for auxiliary task. Can the auxiliary generative model(s) be used for OOD detection? Can you provide its (or their) performance? If the generative model(s) performs worse, what could contribute to the advantage of the predictor (maybe the contrastive learning loss or maybe the auxiliary task)? Can you identify where the advantage comes from?

**Limitations:**

The paper states there is a limitation discussion. Where is it?

---

> ### Author Rebuttal · Authors · 2023-08-09
>
> We sincerely thank you for your constructive comments and generous supports! Please find our responses below.
>
> > Q1. The major concern is the performance of the generator (or possibly I miss that piece of information). While section 3 is convincing, in practice (section 4) how to guarantee high MoG density region has high concentration of in-distribution samples. This may happen when a generator does not perform well.
>
> A1. Sincerely apologize for your confusion. We would like to address your concerns as follows.
>
>
> - **The performance of the generator does not matter.** Our ATOL is different from previous data generation-based methods in that we do not assume the generated data are reliable. Then, even with the unreliable data, they can still benefit OOD detection when it satisfies Conditions 1-2 and Eq. 5. Therefore, we do not need to care about the performance of the generator, making our proposal more attractive than previous works in data generation-based OOD detection.
>
> - **Auxiliary data can differ from real data in the input space.** Generally, any data distribution can benefit our predictor if they can be separated into two disjoint supports, playing the role of auxiliary ID and OOD data, respectively (i.e., Condition 1 should be satisfied). Therefore, if a high MoG density region does not correspond to ID data (which is often the case, as we demonstrated in Figures 4 to 9 in Appendix E.8), it can still benefit our predictor following Proposition 1.
>
> In summary, the key motivation of our ATOL is that: **unreliable data (i.e., auxiliary ID and OOD data) in the input space can still benefit OOD detection, if we align distributions between auxiliary and real data w.r.t. the predictor (i.e., Condition 2).**
>
>
>
> > Q2. Can the author change the term predictor to OOD detector?
>
> A2. As a common practice [1,2], OOD detector is defined by both the predictor and the scoring function. In our paper, we emphasize our focus on the training procedure for the predictor, thus using "predictor" more commonly than "OOD detector". We will further consider the terminologies in our revision, trying to make descriptions clearer.
>
> > Q3. While it is a good idea to generate in-dist and OOD samples using high MoG density region, does it also make the task too easy? I imagine that high MoG region samples may look like in-dist samples, while the rest may look like noise. Does the predictor can perform better if it can learn from more difficult (harder to discern) generated in-dist and generated OOD samples?
>
> A3. Yes. Using hard samples in auxiliary data seems to be a promising way, which may boost detection performance and training efficiency. However, there may still be a lack of a principal way of selecting those data in low density regions, and thus studying how to make the data selection more effective remains an open question. Your constructive suggestions will enlighten us to further improve ATOL, and thanks again for your comments.
>
>
> > Q4. The performance gain of the (OOD) predictor may be comparable to the generative model used for auxiliary task. Can the auxiliary generative model(s) be used for OOD detection? Can you provide its (or their) performance? If the generative model(s) performs worse, what could contribute to the advantage of the predictor (maybe the contrastive learning loss or maybe the auxiliary task)? Can you identify where the advantage comes from?
>
> A4. Sincerely apologize for the confusion. We would like to address your concerns as follows. Related discussions will be added in our revision.
>
>
> - **Generators cannot be used for OOD detection.** In our current realization, the generator is used to generate auxiliary OOD data. It is either randomly initialized or pre-trained on ID data, further satisfying the distance-preserving constraint in Eq. 8. Therefore, the generator in ATOL is not trained particularly for OOD detection, which is hard to be used for OOD detection. However, we also believe that using the generator to detect OOD detection is attractive, which will motivate our following studies.
>
> - **A poor generator can still benefit OOD detection.** Our ATOL is different from previous data generation-based methods in that we do not assume the generator is reliable in generating high-quality data. Especially, in Section 3, we demonstrate that even if the generated data are not reliable, they can still benefit the predictor if Conditions 1-2 and Eq. 5 are satisfied. We also provide the experimental supports in Section 5.3, justifying that a randomly initialized generator can still benefit our method to improve OOD detection. Therefore, the main advantages of our improvements come from the introduced conditions in Section 3 and the learning strategies in Section 4 instead of the adopted generators.
>
>
>
>
>
> > Q5: The paper states there is a limitation discussion. Where is it?
>
> A5. Due to space limitations, our discussion about limitations is given in Appendix F. Here, we list two factors that may motivate our future works. First, our current realization of ATOL is relatively intricate, requiring training constraints on both the generator and the predictor. Further studies will explore more advanced conditions that can ease our realization and further reduce computing costs. Second, we observe that the diversity of generated data is closely related to the final performance (cf., Appendix D.6). However, in our current version, we do not consider diversity for the generator in either theories or algorithms, which will motivate our following exploration.
>
>
> [1]: Weitang Liu, et al. "Energy-based out-of-distribution detection." NeurIPS 2020.
>
> [2]: Du Xuefeng, et al. "Vos: Learning what you don't know by virtual outlier synthesis." ICLR 2022.

---

> > ### Comment · Reviewer_iqYB · 2023-08-11
> > **Thank you for your reply!**
> >
> > Thank you for your reply!
> >
> > First I completely agree with other reviewers that this draft is not very well written because the main points are not clearly answered. I would like to summarize my idea here and confirm with the authors. The main point is that, if the auxiliary generative models could achieve C1 which needs the disjoint support set for gen-in vs gen-OOD, and C2 which needs the gen-in is close to real-in. Then this framework will work.
> >
> > Moreover, based on what the author replies, then how to make sure the sets of gen-in and gen-OOD disjoint while gen-OOD more difficult could make the results better.
> >
> > Can the author(s) confirm the above summary? Thank you!

---

> > > ### Author Response · Authors · 2023-08-13
> > > **Thank you for your follow-up comments!**
> > >
> > > Sincerely thanks for your follow-up comments! Please find our responses below.
> > >
> > > > Q1. First I completely agree with other reviewers that this draft is not very well written because the main points are not clearly answered.
> > >
> > > A1. Sincerely apologize for your confusion. Here, we would like to summarize the motivation of our paper further, hoping that it can help you and other reviewers better understand our proposal.
> > >
> > > 1. **Condition 1: auxiliary ID and OOD data should be disjoint in the data space.**  To make the unreliable generator benefit our predictor in a reliable way, we find that auxiliary ID and OOD data should be disjoint in the data space. In this case, if auxiliary ID data can play the role of real ones w.r.t. the predictor (cf., Condition 2), auxiliary OOD data can also be reliable w.r.t. the predictor (cf., Definition 1). Since these auxiliary OOD data satisfy the standard definition of OOD data following [1], i.e., they have the disjoint support over real ID data. In realization, Eqs. 6-8 are adopted to ensure the disjoint supports between auxiliary ID and OOD data.
> > >
> > > 2. **Condition 2: auxiliary ID data can differ from real ID data in the data space.** As demonstrated in Appendix E.8, generated auxiliary ID data differ from real ID data in semantics/styles. In this situation, our method can still work if the predictor makes no difference between auxiliary and real ID data in their representations (i.e., Condition 2). More extremely, even with the randomly initialized generator, the completely noisy data can still benefit our ATOL (cf., Table 3). In realization, Eq. 9 ensures that the auxiliary ID data are aligned with the real ID data in the representation space of the predictor.
> > >
> > > In summary, **although auxiliary ID and OOD data are not reliable due to the unreliable generator, they can still benefit OOD detection if we can make the predictor "believe" they are reliable**, i.e., Conditions 1-2 and Proposition 1. We will refine our presentation to enhance the readability of our paper in the revision.
> > >
> > > > Q2. I would like to summarize my idea here and confirm with the authors. The main point is that, if the auxiliary generative models could achieve C1 which needs the disjoint support set for gen-in vs gen-OOD, and C2 which needs the gen-in is close to real-in. Then this framework will work.
> > >
> > > A2. Sincerely thank you for the high-level summary of our paper, and your interpretation is completely right. When C1-2 are achieved, we can prove that the Proposition 1 holds. Therefore, even with unreliable OOD sources given by generative models, they can still benefit our models to improve OOD detection.
> > >
> > > > Q3. Moreover, based on what the author replies, then how to make sure the sets of gen-in and gen-OOD disjoint while gen-OOD more difficult could make the results better.
> > >
> > > A3. Yes. Your suggestions are quite insightful, pointing out an important direction that can help us further improve ATOL. It will motivate our following studies, and we sincerely thank you for your constructive comments.
> > >
> > > We will update the related discussion in our revision. We always welcome your new suggestions or comments!

---

> > > > ### Comment · Reviewer_iqYB · 2023-08-18
> > > > **Thank you for your reply!**
> > > >
> > > > Thank you for the confirmation of the main ideas and the resolution of my concerns. I don't have other questions. Thank you!

---

> > > > > ### Author Response · Authors · 2023-08-19
> > > > > **Thank you for your generous supports!**
> > > > >
> > > > > Dear Reviewer iqYB,
> > > > >
> > > > > Glad to hear that your concerns are addressed well. Thanks for your great efforts in reviewing and good questions here.
> > > > >
> > > > > Best regards,
> > > > >
> > > > > Authors of Submission #560

---

### Official Review · Reviewer_eqWq · 2023-07-05

**Soundness:** 4 excellent
**Presentation:** 4 excellent
**Contribution:** 3 good
**Rating:** 7
**Confidence:** 4

**Summary:**

The paper propose to fix the mistaken OOD generation issue in generative model based approach to out-of-distribution data detection, where the mistaken OOD generation means generated OOD data have semantics of ID data. To fix this issue, auxiliary task-based OOD learning (ATOL) is proposed, which is claimed to have the effect of satisfying two key conditions, i.e., auxiliary ID and auxiliary OOD data have disjoint supports in the input space, and auxiliary OOD data are reliable. To achieve this goal, ATOL adds auxiliary task learning loss and ID distribution alignment loss to the real task learning loss. The empirical study shows the non-trivial improvement of OOD detection performance when using ATOL. Ablation study further confirms the effectiveness of each loss in ATOL.

**Strengths:**

The motivation to fix the mistaken OOD generation makes sense to me as shown in Fig. 1.

The two new losses in the proposed ATOL are reasonable and are clearly presented.

I have to say that I like the empirical studies in the paper since they are quite comprehensive and strong, though many interesting results are presented in the supplementary as a result of page limits.

**Weaknesses:**

The drawback and strength of generative model based OOD detection and its comparison with other approaches like scoring or regularized training is not fully discussed in the paper, e.g., performance and efficiency.

In Table 17, the benefit of ATOL is not quite clear when compared with ReAct and CSI.

**Questions:**

1. What is M(z) in Equation 6 and 7? Is it density function of MoG?

2. It is kind of misleading to make the ATOL result bold in Tab. 17 since it is not the best result in terms of AUROC. The best one should be CSI.

3. What is the proportion in Fig. 1b and how is it computed? I tried to search for this information in the paper but it seems that there is no such information. I believe this information is quite important to motivate this paper.

**Limitations:**

Limitations are not discussed in the main paper (please correct me if I am wrong about this).

---

> ### Author Rebuttal · Authors · 2023-08-09
>
> We sincerely thank you for your constructive comments and generous supports! Please find our responses below.
>
> > Q1: The drawback and strength of generative model based OOD detection and its comparison with other approaches like scoring or regularized training is not fully discussed in the paper, e.g., performance and efficiency.
>
> A1: **Data Generation-based Methods vs. Post-hoc Methods**. Post-hoc OOD detection establishes the basis of OOD detection, where various scoring functions in post-hoc methods can be used to improve data generation-based methods. For example, In Appendix E.6, we test ATOL with different scoring strategies, demonstrating that proper scoring functions can lead to improved performance for data generation-based OOD detection. Therefore, post-hoc methods are typically orthogonal to data generation-based approaches.
>
> **Data Generation-based Methods vs. Regularization-based Methods.** Data generation-based OOD detection improves traditional regularization-based methods, in which we do not require real OOD data to fine-tune our predictor. However, due to mistaken OOD generation, previous generation-based methods can make mistakes. It motivates our ATOL to improve conventional data generation-based methods, leading to superior performance over both data generation-based and regularization-based methods.
>
> **Experimental Comparison.** In the main context, our primary focus is on comparing data generation-based approaches. However, we also conduct more extensive evaluations in Appendix E (e.g., Tables 15 to 17), comparing with regularization-based and post-hoc approaches. As we can see, Our ATOL outperforms these approaches, revealing our superiority. Moreover, we also compare the training time for a set of representative methods on CIFAR-100 (similar results on CIFAR-10), summarizing the results in the following table. As we can see, our method demonstrates promising performance improvement over other methods with acceptable computational resources. We will add the related discussion in the Appendix.
>
>
> | Methods | FPR95 | AUROC |Training Time / Epoch|
> | -------- | -------- | -------- | -------- |
> | CSI     | 80.08     | 85.23     | 98.88     |
> | LogitNorm     | 63.45     | 80.18     | **25.16**     |
> | VOS     | 75.41     | 78.20     | 38.97     |
> | NPOS     | 62.72     | 84.17     | 61.54     |
> | ATOL     | 55.22     | 87.24     | 58.33     |
> | ATOL-S     | 43.18    | 88.43     | 70.28     |
> | ATOL-B     | **36.72**     | **91.33**     | 65.33     |
>
>
>
> > Q2: In Table 17, the benefit of ATOL is not quite clear when compared with ReAct and CSI.
>
> A2: In Table 17, we conduct experiments on the ImageNet benchmark, one of the most challenging setups in OOD detection (due to its vast semantic space) [1,2]. Therefore, about $5.55\%$ to $6.28\%$ improvements of our ATOL over CSI and ReAct is relatively promising. How to improve data generation-based OOD detection for tasks with large semantic space remains an open question, which will motivate our future studies.
>
>
>
> > Q3: What is M(z) in Equation 6 and 7? Is it density function of MoG?
>
> A3: Yes, $\mathcal{M}(\mathbf{z})$ in Eqs. 6-7 represents the density function of MoG, namely, $\mathcal{M}(\mathbf{z}) = \sum_{i=1}^c \frac{1}{c} \mathcal{N}(\mathbf{z}|\boldsymbol{\mu}_i, \boldsymbol{\sigma}_i)$, with $\boldsymbol{\mu}_i$ the mean and
> $\boldsymbol{\sigma}_i$ the covariance for the $i$-th sub-Gaussian. We will make it clearer in our revision.
>
> > Q4: It is kind of misleading to make the ATOL result bold in Tab. 17 since it is not the best result in terms of AUROC. The best one should be CSI.
>
> A4. Sincerely apology for our mistakes. We will modify the related parts in our revision.
>
> > Q5: What is the proportion in Fig. 1b and how is it computed? I believe this information is quite important to motivate this paper.
>
> A5. We apologize for the missing description. The proportion in Figure 1(b) is the percentage of ID data mixed in the OOD data, reflecting the severity of wrong OOD data during training. We will describe more about the experimental setups for Figure 1 in the Appendix.
>
> > Q6: Limitations are not discussed in the main paper (please correct me if I am wrong about this).
>
> A6. Due to space limitations, our discussion about limitations is given in Appendix F. Here, we list two factors that may motivate our future works. First, our current realization of ATOL is relatively intricate, requiring training constraints on both the generator and the predictor. Further studies will explore more advanced conditions that can ease our realization and further reduce computing costs. Second, we observe that the diversity of generated data is closely related to the final performance (cf., Appendix D.6). However, in our current version, we do not consider diversity for the generator in either theories or algorithms, which will motivate our following exploration.
>
> [1]: Rui Huang, et al. "MOS: towards scaling out-of-distribution detection for large semantic space." CVPR 2021.
>
> [2]: Dan Hendrycks, et al. "Scaling out-of-distribution detection for real world settings." ICML 2022.

---

> > ### Comment · Reviewer_eqWq · 2023-08-16
> > **After Rebuttal**
> >
> > Thanks for the response, I will keep my score.

---

> > > ### Author Response · Authors · 2023-08-18
> > > **Thanks for supporting our paper to be accepted.**
> > >
> > > Dear Reviewer eqWq,
> > >
> > > We will include these discussions in our revision to improve our submission. We sincerely thank you for supporting our paper to be accepted!
> > >
> > > Best regards,
> > >
> > > Authors of Submission #560

---

### Official Review · Reviewer_3oBY · 2023-07-07

**Soundness:** 3 good
**Presentation:** 2 fair
**Contribution:** 3 good
**Rating:** 5
**Confidence:** 4

**Summary:**

The paper tries to overcome the impact of directly applying incorrect OOD data on the OOD model through auxiliary tasks, thereby improving the performance of OOD tasks. The theoretical part is hard to follow, and the experimental part proves the effectiveness of the theory.

**Strengths:**

1. The paper introduces an auxiliary OOD detection task to combat mistaken OOD generation.
2. The proposed method requires a small additional calculation cost.
3. Experiments show the effectiveness of the proposed method.

**Weaknesses:**

1. The proof of C2 is not clear enough to allow me to clearly understand why achieving C2 can better utilize OOD data. If the model is strong enough or fully trained, it can still confuse incorrect OOD data even if formula 5 is met. Therefore, I judge that the training steps of the algorithm should not be too many and the model should not be too large. Does the author have an explanation for this aspect?

2. There is a writing error in part b of Formula 10.

3. Many aspects of the experiment followed the settings of reference [31], but why not include them in the comparison?

4. The paper is really hard to follow, please polish the paper carefully.

**Questions:**

There are too many Mathematical notations. It is suggested to sort out a table to make it clearer.

**Limitations:**

Yes.

---

> ### Author Rebuttal · Authors · 2023-08-09
>
> We sincerely thank you for your constructive comments and generous supports! Please find our responses below.
>
> > Q1. The proof of C2 is not clear enough to allow me to clearly understand why achieving C2 can better utilize OOD data.
>
> A1. Thank you for your valuable concern. We would like to answer your questions as follows. The related discussions will be added in our revision, especially for Section 3.
>
> - **How to interpret Condition 2?** Measured in the embedding space of the predictor, Condition 2 states that auxiliary ID data approximately follow the same distribution as that of the real ID data. Therefore, a proper predictor should be trained to satisfy such a condition, motivating Eq. 9 in our realization.
>
> - **Why is Condition 2 important?** Auxiliary ID/OOD data can arbitrarily differ from real ID/OOD data in the data space, i.e., auxiliary data are unreliable OOD sources. However, if Condition 2 is satisfied, the model "believes" the auxiliary ID data and the real ID data are the same. Then, the auxiliary OOD data will have the disjoint support over the real ID data and thus achieve reliability w.r.t. the predictor (cf., Proposition 1). Therefore, Condition 2 is critical for our ATOL.
>
>
> > Q2. If the model is strong enough or fully trained, it can still confuse incorrect OOD data even if formula 5 is met. Therefore, I judge that the training steps of the algorithm should not be too many and the model should not be too large. Does the author have an explanation for this aspect?
>
> A2. Many thanks for your question. Conditions 1-2 ensure that unreliable sources will not mislead our predictor in OOD detection. In fact, if the model is stronger, we can better satisfy Condition 2 and Eq. 5. Then, according to Proposition 1, auxiliary OOD data are more likely to be reliable and will not confuse our predictor. The following experimental results on CIFAR can support our above claims, where using a more complex model (i.e., DenseNet-121) can lead to better performance in OOD detection. We will add the related discussion below Proposition 1.
>
> CIFAR-10
> |Method|SVHN||LSUN-Crop||LSUN-Resize||iSUN||Texture||Places365|| Average||
> |:-:|:-:|:-:|:-:|:-:|:-:|:-:|:-:|:-:|:-:|:-:|:-:|:-:|:-:|:-:|
> ||FPR95|AUROC|FPR95|AUROC|FPR95|AUROC|FPR95|AUROC|FPR95|AUROC|FPR95|AUROC|FPR95|AUROC|
> |DenseNet-121|18.05|96.27|1.45|99.58|4.35|99.88|4.45|98.86|20.90|95.70|25.85|94.39|12.51|97.28|
> |WRN-40-2|20.60|96.03|1.48|99.59|5.20|98.78|5.00|98.76|26.05|95.03|27.55|94.33|14.31|97.09|
>
>
> CIFAR-100
> |Method|SVHN||LSUN-Crop||LSUN-Resize||iSUN||Texture||Places365|| Average||
> |:-:|:-:|:-:|:-:|:-:|:-:|:-:|:-:|:-:|:-:|:-:|:-:|:-:|:-:|:-:|
> ||FPR95|AUROC|FPR95|AUROC|FPR95|AUROC|FPR95|AUROC|FPR95|AUROC|FPR95|AUROC|FPR95|AUROC|
> |DenseNet-121|70.30|85.96|61.55|87.93|22.15|95.87|22.30|95.59|48.30|87.87|77.15|79.28|50.29|88.75|
> |WRN-40-2|70.85|84.70|13.45|97.52|51.85|90.12|55.80|89.02|63.10|83.37|75.30|78.86|55.06|87.26|
>
>
> > Q3. There is a writing error in part b of Formula 10.
>
> A3. Thank you for your kind correction. We will correct Eq. 10 in our revision.
>
> > Q4. Many aspects of the experiment followed the settings of reference [31], but why not include them in the comparison?
>
> A4. In our evaluation, we **have conducted** experiments for Free Energy (the method suggested in reference [31] of our paper). We will make it clearer in our revision.
>
>
> > Q5. The paper is really hard to follow, please polish the paper carefully.
>
> A5. Sincerely apologize for your confusion. Following your kind suggestion, we will summarize the key concepts and heuristics in our revision. Besides, we will add the notation table in Appendix and remarks for theories in Section 3, hoping they can help you better understand our paper. If you have any further suggestions, we look forward to discussion and are happy to answer any questions that might arise.
>
> > Q6. There are too many Mathematical notations. It is suggested to sort out a table to make it clearer.
>
> A6. Thank you for the kind suggestion. We summarize the adopted notations in the following tables. We will add the mathematical notations table in the Appendix.
> |Notation|Description|
> |:--:|:--:|
> |**Spaces**||
> |$\mathcal{X}$ and $\mathcal{Y}$|the data space and the label space
> |**Distributions**||
> |$\mathcal{P}\_{\text{X,Y}}^{\text{ID}}$ and $\mathcal{P}\_{\text{X}}^{\text{ID}}$|the joint and the marginal real ID distribution|
> |$\mathcal{P}\_{\text{X}}^{\text{OOD}}$|the marginal real OOD distribution|
> |$\mathcal{G}\_{\text{X,Y}}^{\text{ID}}$ and $\mathcal{G}\_{\text{X}}^{\text{ID}}$|the joint and the marginal auxiliary ID distribution|
> |$\mathcal{G}\_{\text{X}}^{\text{OOD}}$|the marginal auxiliary OOD distribution|
> |$\mathcal{M}\_{\text{Z}}$|the specified MoG distribution|
> |$\mathcal{U}\_{\text{Z}}$ |the specified uniform distribution|
> |**Data**||
> |$\mathbf{x}\_{\text{ID}}$ and $y\_{\text{ID}}$ |the real ID data and label|
> |$\hat{\mathbf{x}}\_{\text{ID}}$ and $\hat{y}\_{\text{ID}}$|the auxiliary ID data and label|
> |$\hat{\mathbf{x}}\_{\text{OOD}}$|the auxiliary OOD data|
> |$\mathbf{z}\_{\text{ID}}$ and $\mathbf{z}\_{\text{OOD}}$|the latent ID data and the latent OOD data|
> |$\mathcal{Z}^{\text{ID}}$ and $\mathcal{Z}^{\text{OOD}}$|the latent ID data sets and the latent OOD data sets|
> |**Models**||
> |$\mathbf{h}$|the predictor: $\mathbb{R}^{n} \rightarrow \mathbb{R}^c$|
> |$\boldsymbol\phi$ and $\boldsymbol\rho$|the feature extractor and the classifier|
> |$s(\cdot;\mathbf{h})$ |the scoring function: $\mathbb{R}^{n}\rightarrow \mathbb{R}$|
> |$f_{\beta}(\cdot)$ | the OOD detector: $\mathbb{R}^{n}\rightarrow \\{\text{ID}, \text{OOD}\\}$, with threshold $\beta$|
> |$G$ |the generator: $\mathbb{R}^m\rightarrow\mathbb{R}^n$|
> |**Loss and Function**||
> |$\ell_{\text{CE}}$ and $\ell_{\text{OE}}$|ID loss and OOD loss|
> |$\ell_{\text{reg}}$|the generator regularization loss|
> |$\ell_{\text{align}}$|the alignment loss|
> |$\boldsymbol{\phi}'(\cdot)$|the mapping function|
> |$\mathcal{M}(\cdot)$|the density function of MoG|

---

> ### Author Response · Authors · 2023-08-13
> **Looking forward to your responses or further suggestions/comments!**
>
> Dear Reviewer 3oBY,
>
> We have addressed your initial concerns regarding our paper (see https://openreview.net/forum?id=87Qnneer8l&noteId=klW9OVRexe). We are happy to discuss them with you in the openreview system if you feel that there still are some concerns/questions. If you have more suggestions, please tell us. We will merge them into our revision as well!
>
> Best regards,
>
> Authors of Submission \#560

---

### Official Review · Reviewer_Q6fc · 2023-07-08

**Soundness:** 3 good
**Presentation:** 3 good
**Contribution:** 3 good
**Rating:** 6
**Confidence:** 5

**Summary:**

One of the techniques for detecting OOD instances is to train a model on OOD data. However, that task is not easy due to difficulty inherent with collecting such OOD data. Rather than collecting such data, this paper proposes instead to generate it, and to train an auxiliary task to improve the OOD detection capabilities of deep learning networks. The proposed approach is called ATOL, for Auxiliary Task-based OOD Learning, and proposes to address one fundamental flaw in existing data-generation based detection methods: the collection of OOD instances from ID data that can mistakenly be labeled as OOD data while being in reality ID.

**Strengths:**

This paper addresses an interesting problem that I think is quite overlooked in the research community. Indeed, many studies focus on collecting OOD instances without necessarily evaluating the possible side effects of the collected data. This paper identified that instances that are collected and labeled as OOD instances can in fact be ID instances, which could lead to poor generalization performance on ID data, and poor OOD detection capabilities. The approach the authors devised to address the problem is sound and quite intuitive, and the evaluation is quite strong.

**Weaknesses:**

Although this paper addresses the OOD detection problem from a data-generation perspective, I would have very much liked to see how their approach fair with other techniques like distance-based OOD detection methods. Some interesting distance-based OOD detection mechanisms have been proposed in the recent past. For instance, CIDER [1] has achieved state-of-the-art performance OOD detection capabilities that shouldn't be overlooked by the researchers approaching OOD detection from a data-generation perspective.  This would help educate mainstream readers more on exactly what techniques to pursue to robustify their models against OOD samples.

A more fundamental limitation of this study is the fact that it heavily relies on a mixture of Gaussian to decide on what latents to consider as OOD and which ones to consider as ID. As MoGs can be sensitive to outliers, have a rather limited expressive power, their accuracy needs to be presented in the paper to showcase their effectiveness in helping collect the data to train the auxiliary task on.

[1]: How To Exploit Hyperspherical Embeddings For OOD detection? https://arxiv.org/pdf/2203.04450.pdf


**Questions:**

Based on the limitations I raised above, I would suggest the authors to perform a comparative study with some of the distance-based OOD detection approaches like CIDER. Additionally, evaluating the performance of the MoGs they used could help validate further the effectiveness of their approach.

**Limitations:**

Yes

---

> ### Author Rebuttal · Authors · 2023-08-09
>
> We sincerely thank you for your constructive comments and generous supports! Please find our responses below.
>
> > Q1. Although this paper addresses the OOD detection problem from a data-generation perspective, I would have very much liked to see how their approach fair with other techniques like distance-based OOD detection methods, e.g., CIDER.
>
> A1. In the main context, we mainly compare our ATOL with other data generation-based approaches. However, we also conduct more evaluations in Appendix E (e.g., Tables 15 to 17), comparing with regularization-based and post-hoc approaches. We include several representative distance-based approaches in these experiments, such as KNN [1] and CSI [2]. As shown in Appendix E, Our ATOL outperforms these approaches, revealing our superiority.
>
> Following your kind suggestions, we further compare with CIDER and ViM [3] (both distance-based methods) on CIFAR-10/100 benchmarks, summarizing the results in the following tables. **Comparing with the SOTA and vital baseline CIDER, our ATOL can still reveal superior results**. We will add the related discussions for CIDER and other distance-based methods to Appendix E in our revision.
>
>
>
> CIFAR-10
> |   Method   |  SVHN |       | LSUN-Crop |       | LSUN-Resize |       |  iSUN |       | Texture |       | Places365 |       | Average |       |
> |:----------:|:-----:|:-----:|:---------:|:-----:|:-----------:|:-----:|:-----:|:-----:|:-------:|:-----:|:---------:|:-----:|:-------:|:-----:|
> |            | FPR95 | AUROC |   FPR95   | AUROC |    FPR95    | AUROC | FPR95 | AUROC |  FPR95  | AUROC |   FPR95   | AUROC |  FPR95  | AUROC |
> |    ViM    | 56.97 | 89.77 |   49.96   | 91.43 |    63.54    | 88.15 | 62.20 | 88.47 |  45.20  | 91.76 |   47.86   | 91.45 |  54.29  | 90.17 |
> |    CIDER    | **5.86** | **98.36** |   7.35   | 98.50 |    47.58    | 93.64 | 47.15 | 93.60 |  28.04  | 94.79 |   41.10   | 91.03 |  29.51   | 94.99 |
> | Our Method | 12.75 | 96.92 |   **4.60**   | **98.92** |    **0.65**    | **99.78** | **0.55** | **99.83** |  **10.25**  | **97.12** |   **22.85**   | **94.80** |  **8.61**  | **97.90** |
>
> CIFAR-100
> |   Method   |  SVHN |       | LSUN-Crop |       | LSUN-Resize |       |  iSUN |       | Texture |       | Places365 |       | Average |       |
> |:----------:|:-----:|:-----:|:---------:|:-----:|:-----------:|:-----:|:-----:|:-----:|:-------:|:-----:|:---------:|:-----:|:-------:|:-----:|
> |            | FPR95 | AUROC |   FPR95   | AUROC |    FPR95    | AUROC | FPR95 | AUROC |  FPR95  | AUROC |   FPR95   | AUROC |  FPR95  | AUROC |
> |    ViM    | 72.32 | 82.92 |   74.03   | 81.57 |    84.89    | 77.03 | 84.15 | 76.69 |  33.51  | 91.72 |   64.17   | 79.57 |  68.78  | 81.58 |
> |    CIDER    | **52.21** | 88.44 |   46.88   | 90.18 |    52.23    | 89.89 | 47.57 | 89.91 |  84.67  | 70.62 |   84.67   | 71.82 |  61.37  | 83.48 |
> | Our Method | 54.65 | **89.69** |   **43.95**   | **91.27** |    **7.80**    | **98.57** | **9.60** | **98.13** |  **37.45**  | **89.51** |   **66.90** | **80.82** |  **36.72**  | **91.33** |
>
>
>
> > Q2. A more fundamental limitation of this study is the fact that it heavily relies on a mixture of Gaussian to decide on what latents to consider as OOD and which ones to consider as ID. As MoGs can be sensitive to outliers, have a rather limited expressive power, their accuracy needs to be presented in the paper to showcase their effectiveness in helping collect the data to train the auxiliary task on.
>
> A2. Sincerely apologize for your confusion. We would like to answer your questions as follows. The related discussion will be added in our revision, especially for Section 3.
>
> - **Reliance on MoG Assumption.** MoG just provides a simple way to generate data, yet the key point is to ensure that auxiliary ID/OOD data should have the disjoint support (i.e., Condition 1). Therefore, if Condition 1 is satisfied properly, other noise distributions, such as the beta mixture models and the uniform distribution, can also be used. We will explore different choices of noise distributions in the future.
>
> - **Fitting the MoG distribution.** Our ATOL is different from previous data generation-based methods in that we do not require generated data to be reliable in the data space. ATOL does not involve fitting the MoG to real ID data, where the parameters can be pre-defined and fixed. Therefore, we do not consider overfitting and accuracy of MoG in our paper.
>
> Although generated data are unreliable in the data space (e.g., auxiliary ID data may differ from its real counterparts), they can still benefit the predictor when Conditions 1-2 and Eq. 5 are satisfied. Heuristically, these conditions and constraints make the predictor take no difference between auxiliary and real ID data. Therefore, auxiliary OOD data are beneficial from the predictor perspective since they have the disjoint support over the real ID data.
>
>
>
> [1]: Yiyou Sun, et al. "Out-of-distribution detection with deep nearest neighbors." ICML 2022.
>
> [2]: Kimin Lee, et al. "A simple unified framework for detecting out-of-distribution samples and adversarial attacks." NeurIPS 2018.
>
> [3]: Haoqi Wang, et al. "Vim: Out-of-distribution with virtual-logit matching." CVPR 2022.

---

> > ### Comment · Reviewer_Q6fc · 2023-08-16
> >
> > I thank the reviewers for the detailed rebuttal and the additional results. The results are compelling. I am raising my rating to Weak Accept.

---

> > > ### Author Response · Authors · 2023-08-18
> > > **Thanks for supporting our paper to be accepted.**
> > >
> > > Dear Reviewer Q6fc,
> > >
> > > Glad to hear that your concerns are addressed well. We will include these experiments in our revision as suggested by you. Thanks for supporting our paper to be accepted!
> > >
> > > Best regards,
> > >
> > > Authors of Submission #560

---

### Decision · Program_Chairs · 2023-09-21

**Decision:**

Accept (poster)

**Comment:**

The paper aims to address the mistaken OOD generation issue (i.e., the generated OOD data are not OOD) in data-generation-based learning methods for OOD detection. The authors proposed to use the generated ID/OOD data as an auxiliary task, not a direct task, to improve the OOD detector for real OOD data. Reviewers generally found the problem important, the proposed approach promising, and the experiments solid.

After the rebuttal, many of the reviewers' concerns were addressed, and several reviewers raised the ratings. The paper now receives a positive rating: average 5.75 (5, 5, 6, 7). The AC thus recommends an acceptance. The AC suggests the authors incorporate their rebuttal responses into the final version and improve parts of the writing/explanations, as requested by the reviewers.